## REPORT

# *TLN1* contains a cancer-associated cassette exon that alters talin-1 mechanosensitivity

Lina M. Gallego-Paez[1]*, William J.S. Edwards[2]*, Manasa Chanduri[3]*, Yanyu Guo[4], Thijs Koorman[5], Chieh-Yu Lee[1], Nina Grexa[1], Patrick Derksen[5], Jie Yan[4,6], Martin A. Schwartz[3,7], Jan Mauer[1,8], and Benjamin Thomas Goult[2]

**Talin-1 is the core mechanosensitive adapter protein linking integrins to the cytoskeleton. The *TLN1* gene is comprised of 57 exons that encode the 2,541 amino acid TLN1 protein. TLN1 was previously considered to be expressed as a single isoform. However, through differential pre-mRNA splicing analysis, we discovered a cancer-enriched, non-annotated 51-nucleotide exon in *TLN1* between exons 17 and 18, which we refer to as exon 17b. TLN1 is comprised of an N-terminal FERM domain, linked to 13 force-dependent switch domains, R1-R13. Inclusion of exon 17b introduces an in-frame insertion of 17 amino acids immediately after Gln665 in the region between R1 and R2 which lowers the force required to open the R1-R2 switches potentially altering downstream mechanotransduction. Biochemical analysis of this isoform revealed enhanced vinculin binding, and cells expressing this variant show altered adhesion dynamics and motility. Finally, we showed that the TGF-β/SMAD3 signaling pathway regulates this isoform switch. Future studies will need to consider the balance of these two TLN1 isoforms.**

## Introduction

Integrin-dependent adhesion of cells to the extracellular matrix (ECM) not only mediates cell attachment and physical integrity but is also critical in sensing the composition, organization, and mechanical properties of the ECM. There are two talin genes in mammals, widely expressed talin-1 (*TLN1*) and more specialized talin-2 (*TLN2*; Monkley et al., 2001; Senetar and McCann, 2005). Both talins bind integrin beta subunit cytoplasmic domains and directly link them to F-actin and thus are essential for integrin-mediated adhesion (Calderwood et al., 2013; Klapholz and Brown, 2017). *TLN1* and *TLN2* have identical domain structure (Debrand et al., 2009; Gough and Goult, 2018), comprising an N-terminal head domain followed by an 80-residue unstructured linker and then a series of 13, 4- or 5- helical bundles designated rod domains 1–13 (R1–R13; Goult et al., 2013). These helix bundles act as mechanochemical switches that can open and close independently in response to mechanical force. Distinct protein partners bind specifically to helix bundles in the open or closed states such that domain opening and closing controls ligand binding (Goult et al., 2021, 2018). Force-dependent recruitment of signaling molecules thus enables talin to act as a mechanosensitive signaling hub between the ECM

and the cell (Goult et al., 2018). As a result, talin plays a central role in cellular mechanosensing of ECM properties or applied strains. Talin-2 shows tissue-specific alternative splicing of the complex *TLN2* gene (Debrand et al., 2009). However, in studies to date, talin-1 has been assumed to exist as a single constitutive isoform.

Bioinformatic pre-mRNA splicing analysis of pan-cancer RNA-Seq data sets from The Cancer Genome Atlas (TCGA) revealed a robustly transcribed undocumented mRNA sequence in *TLN1* (Gallego-Paez and Mauer, 2022). This region contains an additional non-annotated exon, located between exons 17 and 18, which we refer to as exon 17b. Sequence analysis revealed that exon 17b codes for an in-frame insertion of 17 amino acids into the first helix of the R2 helical bundle. Although exon 17b can be readily detected in many healthy tissues including skin and pancreas, this novel *TLN1* exon is significantly enriched in certain molecular cancer subtypes. Moreover, the presence of *TLN1* exon 17b correlates with altered drug responses and changes in gene dependencies in cancer cell lines suggesting a role in cancer physiology. In this report, we characterized this novel splice variant of talin and showed that the *TLN1* gene is more complex

..........................................................................................................................................................................
[1]BioMed X Institute (GmbH), Heidelberg, Germany; [2]School of Biosciences, University of Kent, Canterbury, UK; [3]Departments of Internal Medicine (Cardiology) and Yale Cardiovascular Research Center, New Haven, CT, USA; [4]Mechanobiology Institute, National University of Singapore, Singapore, Singapore; [5]Department of Pathology, University Medical Center Utrecht, Utrecht, Netherlands; [6]Department of Physics, National University of Singapore, Singapore, Singapore; [7]Departments of Cell Biology and Biomedical Engineering, Yale School of Medicine, New Haven, CT, USA; [8]Department of Immunology, Novartis Institutes for BioMedical Research, Basel, Switzerland.

*L.M. Gallego-Paez, W.J.S. Edward, and M. Chanduri contributed equally to this paper.   Correspondence to Benjamin Thomas Goult: b.t.goult@kent.ac.uk;   Jan Mauer: jan.mauer@novartis.com.

than originally thought. Inclusion of the 17b cassette exon into *TLN1* mRNA provides a previously unrecognized mechanism for the cell to alter mechanotransduction in response to signals.

## Results

### Discovery of a novel exon in *TLN1*

We recently developed a splicing analysis pipeline, DJExpress, that uses junction expression information from RNA sequencing data to identify alternative splicing events in a transcriptome-wide manner (Fig. 1 A; Gallego-Paez and Mauer, 2022). Notably, one of the features of DJExpress is the quantification of both annotated and non-annotated junctions. Thus, DJExpress supports the discovery of undocumented splice events.

Analysis of differential junction expression in TCGA data across all cancers identified such a non-annotated splice event in *TLN1* mRNA. Two novel junctions in *TLN1* mRNA, 18 and 20, were significantly increased in several TCGA cancer patient cohorts compared to healthy control tissue from the Genotype-Tissue Expression (GTEx; Lonsdale et al., 2013) database (Fig. 1 B). In parallel, we detected significantly reduced expression of the previously thought constitutive junction 19 of *TLN1* mRNA (Fig. 1 B). These three junctions are located within the TLN1 coding region (Fig. S1 A). Utilization of junctions 18 and 20 leads to the inclusion of an in-frame 51-nucleotide sequence flanked by exons 17 and 18, which we term exon 17b (Fig. 1 C). Upon inclusion, 17 amino acids are inserted between TLN1 R1 and R2.

### TLN1 exon 17b is a natural splicing event present in healthy tissues and enriched in certain cancer subtypes

We initially discovered *TLN1* exon 17b through analysis of differential splicing events in cancer tissues (Gallego-Paez and Mauer, 2022). Nevertheless, we readily detected *TLN1* exon 17b expression in normal tissues by analysis of GTEx RNA-Seq data sets (Fig. 1 D). Although expressed at low to non-detectable levels in many tissues, some tissues such as skin and pancreas show high expression of exon 17b and others including kidney cortex, endocervix, testis, pituitary, liver, and spleen show robust expression as well. These data suggest that exon 17b is likely relevant to normal physiology. Interestingly, there was evidence for exon 17b expression in some primates but exon 17b was to date not annotated in humans (Fig. S1 B).

We next addressed whether *TLN1* exon 17b splicing, in addition to primary cells, can also be detected in cancer cell lines. Analysis of expression data from the Cancer Cell Line Encyclopedia (CCLE; Barretina et al., 2012) revealed that exon 17b status varied within cancer subtypes including breast, lung, and colon cancer (Fig. 1 E). Notably, exon 17b is enriched in breast, cervix, colon, and lung cancer as indicated by reduced expression of the exon 17b-skipping junction 19 and increased expression of exon 17b inclusion junctions 18 and 20 (Fig. S1 C). Moreover, in PAM50 breast cancer molecular subtypes, we found TLN1 exon 17b inclusion to be lowest in normal-like and highest in basal-like tumors (Fig. 1 F). These data suggest that TLN1 alternative splicing is part of the phenotypic differences between these cancer subtypes.

Analysis of cancer cell lines across lung, colon, and breast (Fig. 2 A) indicated that exon 17b usage varied between different lines with breast cancer cell lines clustered into two distinct groups, one group with almost 100% spliced in (a value of 1.0 PSI) and another where the PSI was ~0.0. To independently validate splicing of exon 17b, we next performed RT-PCR on cDNA generated from two breast cancer cell lines with high exon 17b inclusion (BT20, ZR751) and two breast cancer cell lines with low 17b inclusion (MDA231, BT549), according to *DJExpress* analysis (Fig. 2 A). Indeed, exon 17b-specific RT-PCR of high exon 17b cell lines resulted in a 234 bp amplicon, indicating exon 17b inclusion (Fig. 2 B). In contrast, cell lines with a predicted low expression of exon 17b produced a 183 bp amplicon, which reflects exon 17b skipping (Fig. 2 B). Furthermore, we tested the inclusion of exon 17b in a panel of 2D and 3D grown cancer cell lines which we had available in the lab (Fig. S1 D) and here we clearly detected both forms in 2D grown cells, but in 3D grown cells we exclusively saw the 17b version. The presence of *TLN1* exon 17b in cancer cell lines can thus be detected by conventional RT-PCR, reflected by a clear shift in amplicon size, without the need for RNA-Seq analysis.

Together these data demonstrate the sensitivity of the *DJExpress* pipeline for robustly identifying and classifying differential splicing events. Moreover, our finding that TLN1 exon 17b is present in some cancer cell lines but absent in others suggests that TLN1 alternative splicing might alter cancer cell behavior, and whose level might be useful for stratification of certain molecular subtypes of breast cancer.

Following our detection of this additional talin-1 exon, we reported our finding to the Ensembl team who verified the presence of a missing cassette exon at chr9:35718356-35718406 (−), having extensive transcriptional support and being constrained as protein-coding in mammals. As a result, this exon was added to the new model ENST00000706939, which was included in Ensembl 109 (released February 8, 2023) and UniProt (accession number A0A1S5UZ07).

### TLN1 exon 17b is associated with altered drug responses and gene dependencies in cancer cell lines

We next asked whether exon 17b might functionally impact on cancer cell physiology. For this, we first tested whether exon 17b inclusion correlates with drug responses. Comparison of DepMap cancer cell line drug sensitivity data (Meyers and Bryan, 2017; Corsello et al., 2020) with exon 17b inclusion and exclusion junction expression revealed that exon 17b inclusion correlates with increased sensitivity to EGFR inhibitors, represented by a negative correlation with drug area under the curve (AUC; Fig. 2 C). Conversely, cell lines that express TLN1 exon 17b were resistant to drugs targeting PI3K signaling and cytoskeleton regulation (Fig. 2 C). Thus, exon 17b inclusion correlates with responses to a variety of drugs, suggesting that exon 17b status may support prediction of therapeutic success.

Therapeutic vulnerability can arise from an increased dependency on the expression of certain genes in cancer cells (Tsherniak et al., 2017). Since TLN1 exon 17b inclusion was associated with altered drug sensitivity, we reasoned that a similar association could also be detected for genetic dependencies. Thus, we next tested whether alternative splicing of *TLN1* is associated with altered gene dependencies in functional genomic

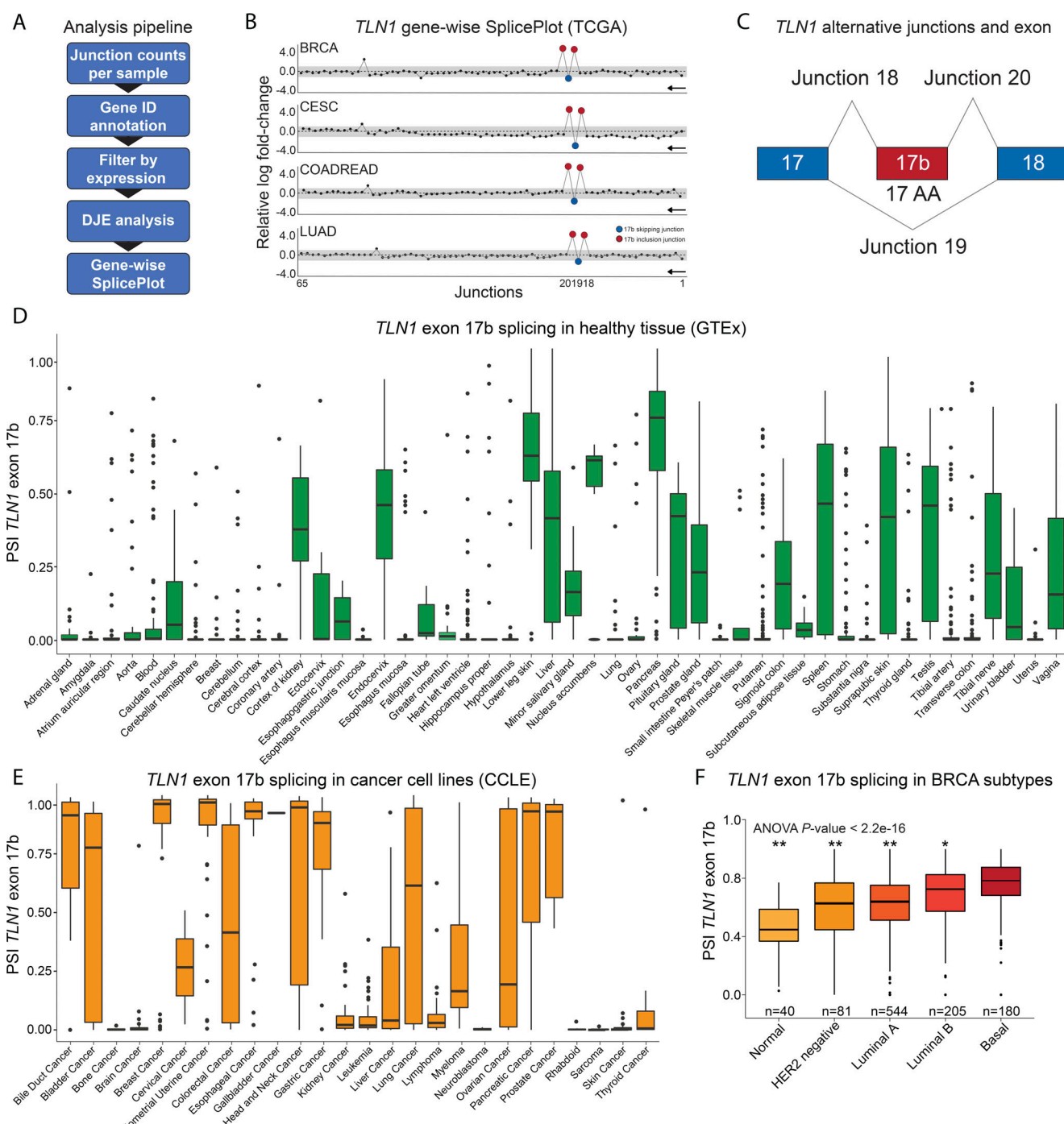

Figure 1. **Differential splicing of a non-annotated *TLN1* cassette exon. (A)** The schematic shows the workflow of the *DJExpress* differential splicing analysis pipeline (DJE = differential junction expression). **(B)** Discovery of a novel cassette exon in TLN1 through differential splicing analysis of cancer patients and healthy controls. Gene-wise SplicePlots of *TLN1* mRNA generated with the *DJExpress* pipeline facilitated identification of two differentially expressed non-annotated junctions indicating the presence of an exon inclusion event (junctions 18–20) between exon 17 and 18 in several cancer types. Representative SplicePlots from breast invasive carcinoma (BRCA), cervical squamous cell carcinoma (CESC), colon and rectal adenocarcinoma (COADREAD), and lung adenocarcinoma (LUAD) cancer tissue from the Cancer Genome Atlas (TCGA) vs. normal tissue from the Genotype-Tissue Expression (GTEx) database are shown. The black arrow indicates the direction of *TLN1* transcription on the reverse strand. Numbers on the x-axis indicate the first, last, and differentially expressed junctions in the gene. Circles represent all junctions that were detected across the full-length TLN1 transcript. Significantly down- and upregulated junctions with |logFC| above cut-off and FDR < 0.05 are shown in blue and red, respectively. Junctions with FDR > 0.05 for absolute or relative logFC (or both) are shown in black. Gray area indicates the log-foldchange cut-off (|logFC| > 1.0). **(C)** Schematic showing the alternatively spliced region of *TLN1*. The constitutive exons 17 and 18 are shown in blue. The presence of junctions 18 and 20 indicate the inclusion of the 17 amino acid (AA) cassette exon 17b (red) whereas the presence of junction 19 indicates exon 17b exclusion. **(D–F)** The non-annotated exon 17b in *TLN1* mRNA has variable expression levels in healthy tissues and is utilized across cancer cell lines. **(D)** Boxplots showing percent spliced-in (PSI) values of *TLN1* exon 17b across GTEx healthy tissues. PSI values range from 0.0–1.0

where a value of 1.0 represents 100% spliced-in. Although absent in many healthy tissue types, exon 17b is highly enriched in some tissues including skin and pancreas. **(E)** Boxplots showing PSI values of *TLN1* exon 17b across cancer cell lines (CCLE). **(F)** Boxplots showing PSI values of *TLN1* exon 17b across PAM50 subtypes of TCGA BRCA breast cancer tumor samples (* P value ≤ 0.01; ** P value ≤ 0.0001 vs. basal subtype, one-way ANOVA).

screens. For this, we used pan-cancer cell line CRISPR screen data from the DepMap project and correlated exon 17b expression with gene effect. We then performed gene set enrichment analysis across all genes that significantly correlated with *TLN1* exon 17b inclusion. Here, we observed enrichment of pathways related to PI3K signaling, cytoskeleton organization, focal adhesion, and EGFR/ErbB signaling correlated with exon 17b expression (Fig. 2 D). These data suggest that exon 17b status may differentially affect genetic dependencies in cancer cells and support the notion that exon 17b status alters cancer cell physiology. Notably, the enriched dependency pathways reflect the same pathways that are targeted by the drugs which we associated with 17b status, thus further supporting the notion that TLN1 alternative splicing links to changes in cancer cell behavior.

### TLN1 exon 17b is regulated by TGF-β/SMAD3 signaling

Since nothing is known about the regulation of *TLN1* alternative splicing, we next wanted to identify regulators of exon 17b inclusion. For this, we re-analyzed published deep RNA-Seq data sets with DJExpress and assessed exon 17b status under different conditions. Re-analysis of a study that examined alternative splicing in HeLa cells, provided clear evidence of exon 17b status regulation (Tripathi et al., 2016). At baseline, HeLa cells show exon 17b inclusion, however stimulation with a combination of TGF-β and EGF lead to a clear splicing switch, which resulted in exon 17b skipping (Fig. 2 E). Moreover, knockdown of the TGF-β downstream transcription factor SMAD3 blocked TGF-β/EGF-induced exon 17b skipping (Fig. 2 E). Similar effects were observed after knockdown of the RNA binding protein PCBP1 (Fig. 2 E). These results strongly suggest that the TGF-β-SMAD3-PCBP1 pathway regulates exon 17b splicing. Notably, TGF-β signaling is a well-described target in cancer therapy (Liu et al., 2021) and some of the beneficial effects of TGF-β inhibition drugs might be contributed to by changes in exon 17b status.

We next asked whether *TLN1* alternative splicing is correlated with other splicing events dysregulated in cancer. For this, we revisited our transcriptome-wide alternative splicing analysis in cancer vs. healthy tissue. Here, we found that splicing of exon 11 in Calsyntenin 1 (*CLSTN1*) negatively correlated with *TLN1* exon 17b splicing in lung and breast cancer patients and cell lines (Fig. S1 D and Fig. S2 A), suggesting that these two splice events might be "trans-mutually exclusive." This exclusivity was also reflected by RT-PCR analysis of breast cancer cell lines where cell lines with *TLN1* exon 17b inclusion exhibited *CLSTN1* exon 11 skipping and vice versa (Fig. S1 D and Fig. S2 B). Notably, *CLSTN1* exon 11 splicing exhibited the same dynamic, but inverse regulation upon TGF-β/EGF stimulation as TLN1 exon 17b (Fig. S2 C). These data indicate that TLN1 and CLSTN1 mRNA might be part of the same TGF-β/EGF-dependent alternative splicing program, which may be subject to future studies.

### Exon 17b alters the primary sequence of talin-1

We next addressed the possible impact of exon 17b on TLN1 protein function. TLN1 is comprised of 2,541 amino acids, that make up the 18 talin domains (Fig. 3 A). Exon 17b introduces a 17 amino acid, in-frame insertion immediately after residue Gln665 (Fig. 3 B). Gln665 is in the first helix of the R2 bundle (Fig. 3, B–D). The crystal structure of talin R1R2 has been solved, which revealed that the two rod domains pack against each other in a side-to-side arrangement (Fig. 3 C; Papagrigoriou et al., 2004). To establish the effect of 17b on this region, we generated structural models of both R1R2-WT and the R1R2-17b proteins using the protein structure prediction tool AlphaFold (Senior et al., 2020). Both the R1R2-WT and R1R2-17b structural models (Fig. S1 E) showed good agreement with the crystal structure of the wild-type R1R2 (Papagrigoriou et al., 2004) validating that the models were accurate. The R1R2-17b structural model showed that the 17 amino acid insert extends the linker region between R1 and R2 (Fig. 3, E and F) and is predominantly unstructured except for a small helical region in the linker (Fig. 3 F), this helical region is from residues that were originally part of R2 before the insertion. The model also predicts structural differences within both the R1 and R2 domains. With insertion of the 17 aa sequence, the first helix of R2 is one turn shorter because the linker now incorporates the first three residues Pro662-Gln665, from the helix. Furthermore, in the wild-type R1R2 structure the linker between R1 and R2 is short, tightly linking the end of R2 with the start of R1 (Fig. 3 D). The extended linker relieves this conformational constraint so that the last helix of R1 is extended by three residues (E657, S658, D659; Fig. 3 F). Interestingly, this extension comes from residues from in the R1R2 linker (blue) that, with the increased linker length, become part of the fifth helix of R1. It seems reasonable to assume that the helix could be extended in a similar fashion in WT-TLN1 if R2 was to unfold.

Full-length TLN1 can adopt open (Fig. 3 A) and autoinhibited (Fig. 3 G) states; overlaying the structural model of R1R2-17b on the cryo-EM structure of autoinhibited monomeric TLN1 (Dedden et al., 2019) shows that the 17b insertion is exposed on the surface of the closed conformation. In the context of the 250 kD full-length molecule, these additional 17 residues will be hard to detect by SDS-PAGE, which probably contributed to its late identification. In the context of R1R2 alone, the splice variant is readily detectible by SDS-PAGE due to the additional ~2 kD (Fig. 3 H). Together, these data suggest that the 17b insertion may perturb the mechanical behavior and function of the R1R2 region of talin1.

### Exon 17b alters the biophysical and biochemical properties of R1R2

To investigate the biophysical and biochemical impacts of the 17-residue insertion on the R1R2 structure, we generated expression constructs, termed R1R2-WT and R1R2-17b. Both proteins expressed well and were amenable to characterization (Fig. 3 H).

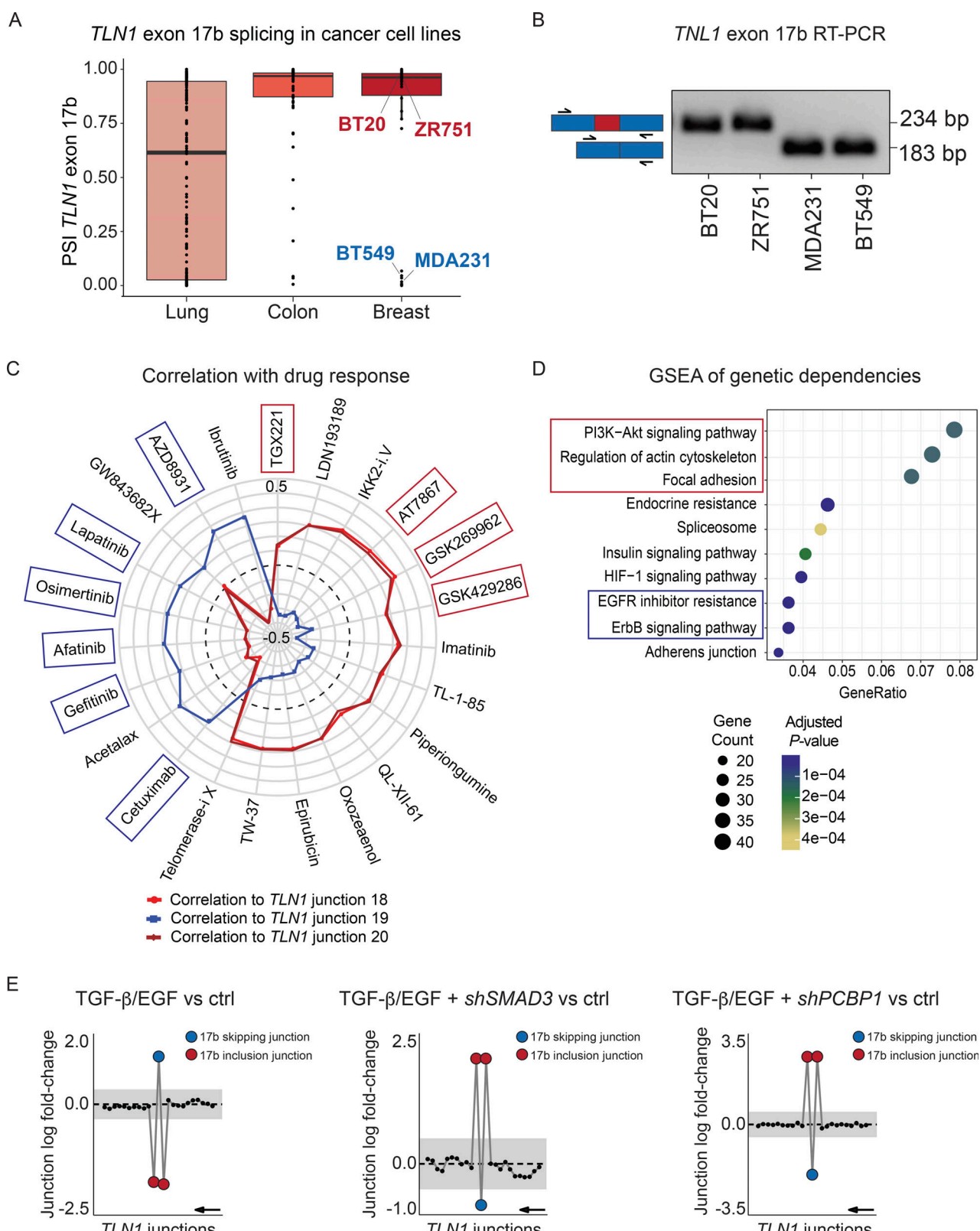

Figure 2. **TLN1 exon 17b inclusion is associated with altered drug response and gene dependencies in cancer cell lines. (A)** Differential splicing of *TLN1* exon 17b is detected in cancer cell lines. Boxplots show distribution of percent spliced-in (PSI) values for *TLN1* exon 17b in lung, colon, and breast cancer cell lines. The individually labeled BT20, ZR751, MDA231, and BT549 breast cancer cell lines were used for RT-PCR validation in B. **(B)** RT-PCR validation of *TLN1* exon 17b expression in four representative breast cancer cell lines. RT-PCR was performed with primers flanking exon 17b (primer positions indicated by black arrows). Exon 17b spans 51 base pairs (bp) and exon 17b inclusion results in an amplicon size increase from 183 bp to 234 bp. BT20 and ZR751 cell lines show exon 17b inclusion whereas MDA231 and BT549 cell lines show exon 17b skipping. **(C)** Expression of *TLN1* inclusion junctions (18, 20) and skipping junction (19)

was correlated to cell survival after drug treatment using DepMap drug sensitivity data across all cancer cell lines. The top-ranked correlation coefficients (FDR < 0.05 and |rho| > 0.2) were used to construct the SpliceRadar plot. *TLN1* exon 17b inclusion (red and dark red lines) and exclusion (blue line) junction expression is plotted against their correlation coefficient with cell survival upon drug treatment. The data suggest that exon 17b inclusion is associated with increased sensitivity to EGFR inhibitors (blue boxes) and resistance to drugs targeting PI3K-Akt and cytoskeleton organization (red boxes). The black dashed line indicates a correlation coefficient $R = 0$ and an $R$ range from −0.5 to 0.5 is shown. **(D)** KEGG gene set enrichment analysis (GSEA) of DepMap gene dependencies associated with *TLN1* exon 17b inclusion in cancer cell lines. The enrichment plot shows the top over-represented pathways, including cell adhesion, cytoskeleton organization (red), and EGFR/ErbB signaling pathways (blue). Dot size represents the number of genes enriched in each KEGG pathway and the color gradient indicates significance level of adjusted P-values. **(E)** Combined TGF-β/EGF treatment promotes TLN1 exon 17b skipping in a SMAD3 and PCBP1-dependent manner. Gene-wise splice plots of TLN1 junction expression in HeLa cells, which show baseline inclusion of exon 17b. Left panel: Combined TGF-β/EGF treatment leads to exon 17b skipping in HeLa cells. Middle panel: shRNA-mediated knockdown of TGF-β signal transducer SMAD3 blocks the TGF-β/EGF-induced skipping of exon 17b in HeLa cells. Right panel: shRNA-mediated knockdown of the RNA-binding protein PCBP1 blocks the TGF-β/EGF-induced skipping of exon 17b in HeLa cells. (The plots shown in this figure were generated by DJExpress-based re-analysis of RNA-Seq data from GSE72419; gray area indicates the log-fold change cut-off (|logFC| > 0.5). Exon 17b inclusion junctions are shown in red, exon 17b skipping junction is shown in blue. Junctions with FDR > 0.05 for absolute or relative logFC (or both) are shown in black. Black arrow indicates the direction of *TLN1* transcription on the reverse strand.

Far-UV CD spectra of R1R2-WT and R1R2-17b confirmed that both constructs were predominantly alpha-helical as predicted by structural analysis (Fig. 3 and data not shown). However, CD thermostability analysis, where the CD is measured at a fixed wavelength (222 nm) over a range of temperatures revealed a significant difference in the stabilities of the two proteins (Fig. 4 A). R1R2-WT showed a single transition confirming a cooperative unfolding event where both R1 and R2 helical bundles unfold simultaneously at a melting temperature ($T_m$) of 77°C. In contrast, R1R2-17b showed similar cooperative unfolding but a markedly reduced $T_m$ of 64°C. These data reveal that the two switch domains R1 and R2 are both folded but destabilized by the 17b insertion.

### Impact of 17b on the interactions of R1R2 with RIAM and vinculin
We next wanted to assess the effect of 17b inclusion on the interactions of R1R2 with ligands. The Rap1-effector RIAM binds to the folded R2 domain (Goult et al., 2013), while vinculin can bind to three vinculin binding sites (VBS); R1 contains a single VBS on helix 4, and R2 contains two VBS, in helices 1 and 4 of the four-helix bundle, that are only available after mechanical force induces domain unfolding, thus, fully unfolded R1R2 can bind up to three vinculins (Gingras et al., 2005; Yao et al., 2014).

The stability of the bundle in which a VBS is embedded is critical as the domain needs to unfold to enable vinculin to bind, suggesting that the altered stability we observed via CD (Fig. 4 A) might alter the VBS availability. To test how 17b affected vinculin binding we used analytical gel filtration where the talin proteins are preincubated with vinculin D1 domain (VD1) prior to loading on the size exclusion column. In line with earlier studies, only modest VD1 binding was seen with R1R2-WT after incubation at 37°C (Fig. 4 B and Fig. S1 F; Papagrigoriou et al., 2004). However, substantially more complexation with R1R2-17b was observed, with both 1:2 and 1:3 talin/vinculin complex peaks detected. These data show that introduction of 17b reduces the stability of the R1R2 region and enhances vinculin binding.

We next used NMR to assess RIAM binding by collecting NMR spectra of R1R2 alone, or in the presence of a threefold excess of RIAM peptide. Comparison of the 1H,15N TROSY spectra of the R1R2-WT and R1R2-17b showed both versions had similar peak dispersion, line widths, and peak positions, confirming that the core folding of the two domains was the same.

Upon addition of a threefold excess of RIAM, extensive chemical shift changes, predominantly to the peaks from the R2 domain, were observed (Fig. 4 C), indicative of binding as shown previously (Goult et al., 2013). Similar spectral changes were observed with R1R2-17b (Fig. 4 D) indicating that the R1R2-17b variant binds similarly confirming that the LD-motif binding surface on R2 is intact.

### The effect of exon 17b on the mechanical response of talin
Talin mediates force transmission from ECM-bound integrin to cytoskeleton. To test whether 17b inclusion alters the mechanical stability of R1 and R2, we performed single molecule experiments using our in-house developed magnetic-tweezer setup (Chen et al., 2011). In this assay, the protein is tethered between the microscope coverslip and a paramagnetic microbead and magnetic tweezers are used to apply forces which increase with time at a fixed loading rate of 1 pN/s. The forces at which mechanical unfolding of the domains occurred (stepwise bead height increases in Fig. 4, E and F) are recorded over multiple cycles and with multiple tethers. For the single molecule experiments we used the R1-R3 triple domain construct as the mechanical response of this protein has been extensively characterized (Yao et al., 2014; Yao et al, 2016).

Both R1-R3-WT and R1-R3-17b show three unfolding events, with both having unfolding of a domain around the same force peaked at ~5 pN (Fig. 4, E and F), which we assign to unfolding of the R3 domain based on its highly characteristic signature reported in our previous studies (Yao et al., 2014; Yao et al., 2016). The unfolding of R2 and R1 domains occurs at higher force, which in the R1-R3-WT are the unfolding events at ~15 and 25 pN (Fig. 4 E). Strikingly, the introduction of 17b results in a marked reduction of the force required to unfold R2 and R1, which peaked at ~13 and ~21 pN (Fig. 4 F). Such a decrease by a few pN is significant since each unfolding step is coupled to large unfolding step sizes (>20 nm). Overall, these results indicate that 17b inclusion results in a reduced mechanical stability of R1 and R2.

### Cells expressing talin1-17b have altered adhesions
To determine whether exon 17b affects TLN1 function in cells, *Tln1⁻/⁻* mouse embryonic fibroblasts were transfected with full-length TLN1 WT (Kumar et al., 2016) or full-length TLN1 exon 17b isoform. This analysis was done with and without depletion

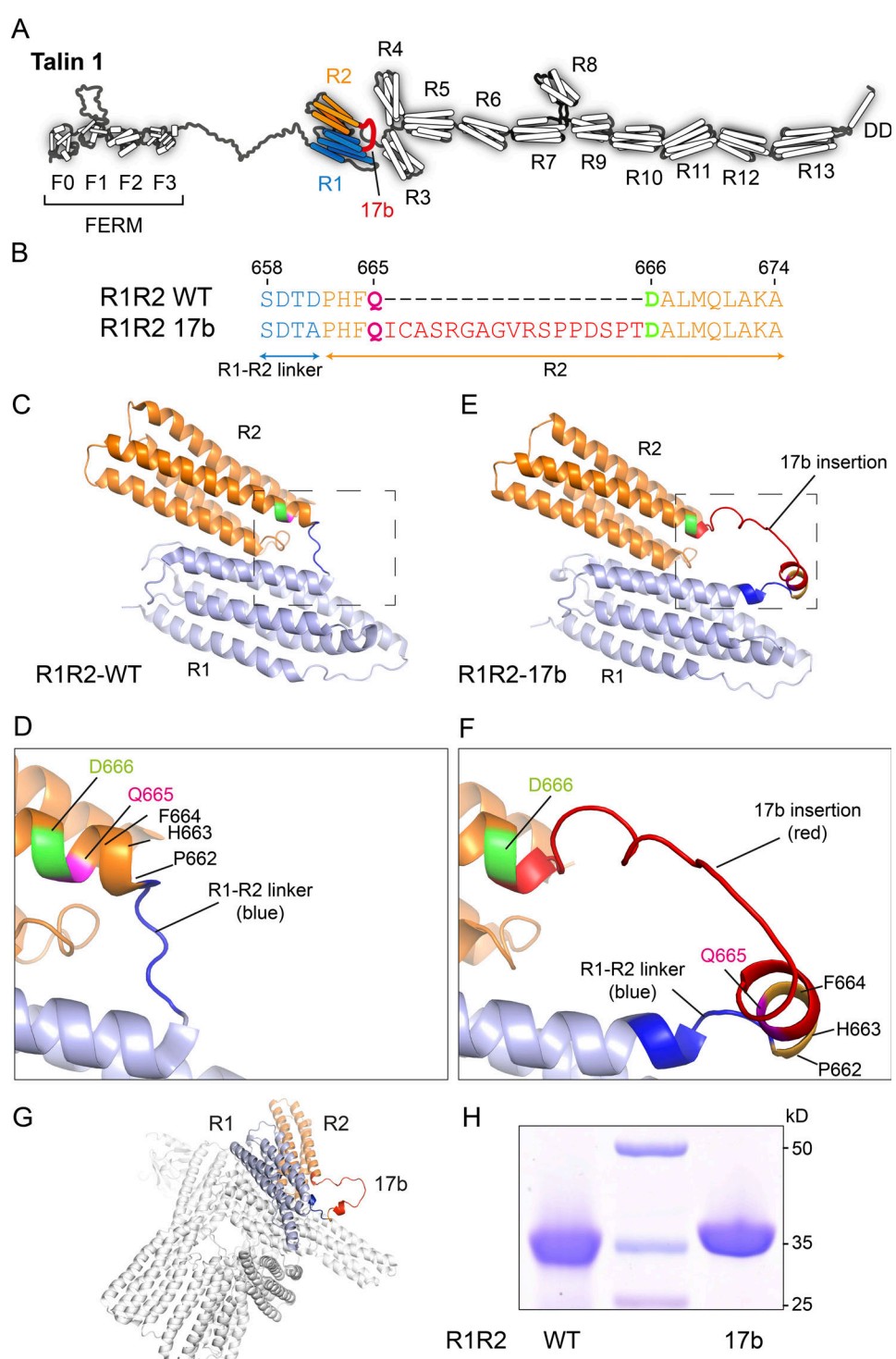

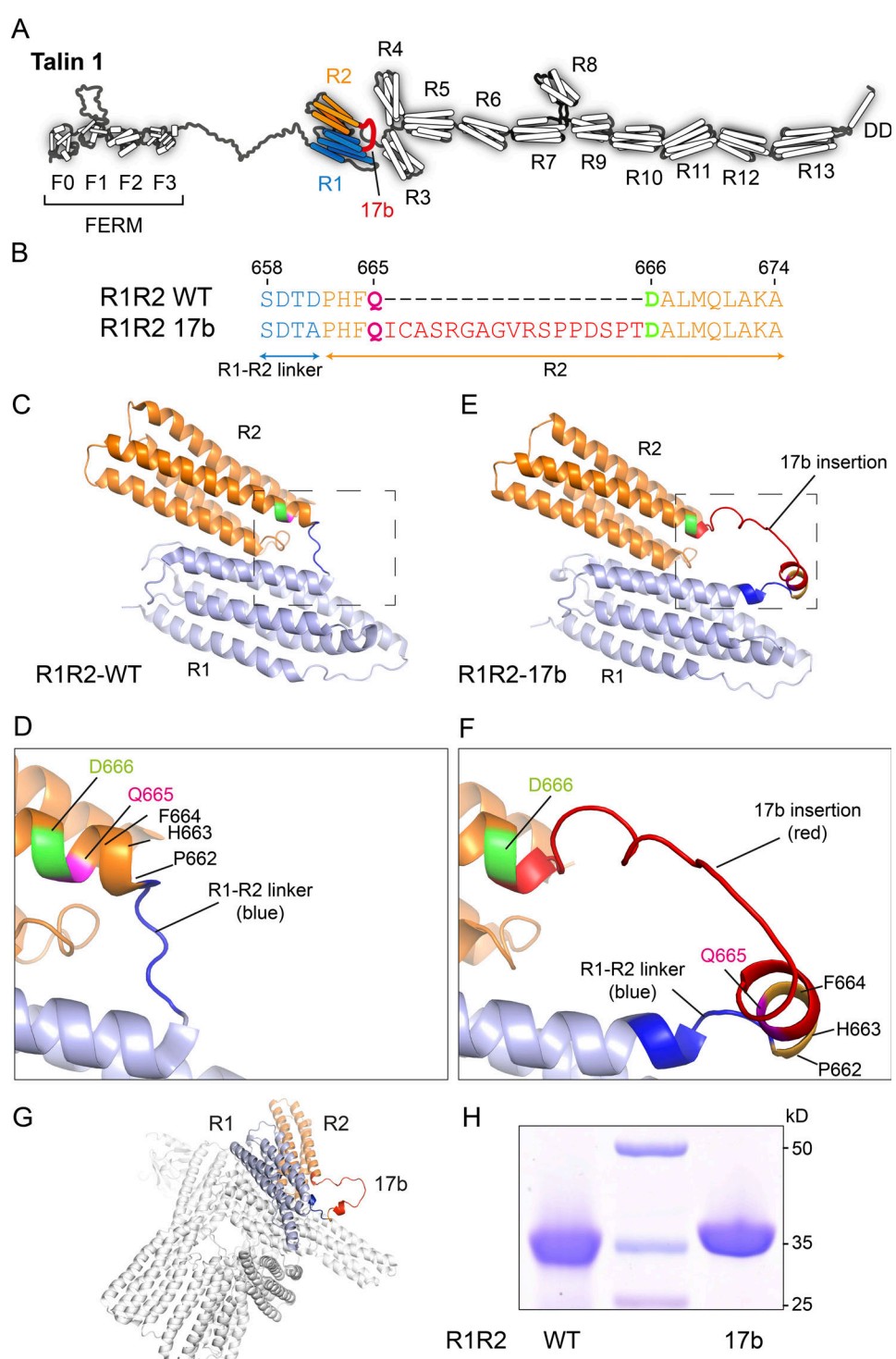

**Figure 3. Structural analysis of exon 17b at the protein level. (A)** Cartoon showing the location of the 17b insertion (red) in the TLN1 protein between R1 (blue) and R2 (orange). **(B)** Sequence alignment of the region of R1R2-WT and R1R2-17b, showing the 17-residue insertion (red) in the R2 region (orange), the residues either side of the insertion, Q665 and D666, are highlighted in magenta and green, respectively. **(C and D)** Crystal structure of R1R2-WT (Protein Data Bank accession no. 1SJ8). The same color scheme as in B is used with R1 (light blue), R1-R2 linker (blue), and R2 (orange). **(D)** Zoomed in region of the R1-R2 linker (blue) showing the location of the insertion site between Q665 (magenta) and D666 (green) in the first helix of R2. **(E and F)** AlphaFold model of R1R2-17b. **(E)** The 17b insertion is shown in red. **(F)** Zoomed in view of the same region as in D. Residues on either side of the 17b insertion, Q665 (magenta) and D666 (green) are highlighted. The last helix of R1 is extended to contain part of the blue linker region, but the first helix of R2 is shortened. **(G)** Cryo-EM structure of full-length TLN1 (Protein Data Bank accession no. 6R9T [Dedden et al., 2019]) overlaid with the structural model of R1R2-17b, indicates that the insertion extends out of the autoinhibited monomeric state. **(H)** SDS-PAGE of the R1R2-WT and R1R2-17b recombinant proteins.

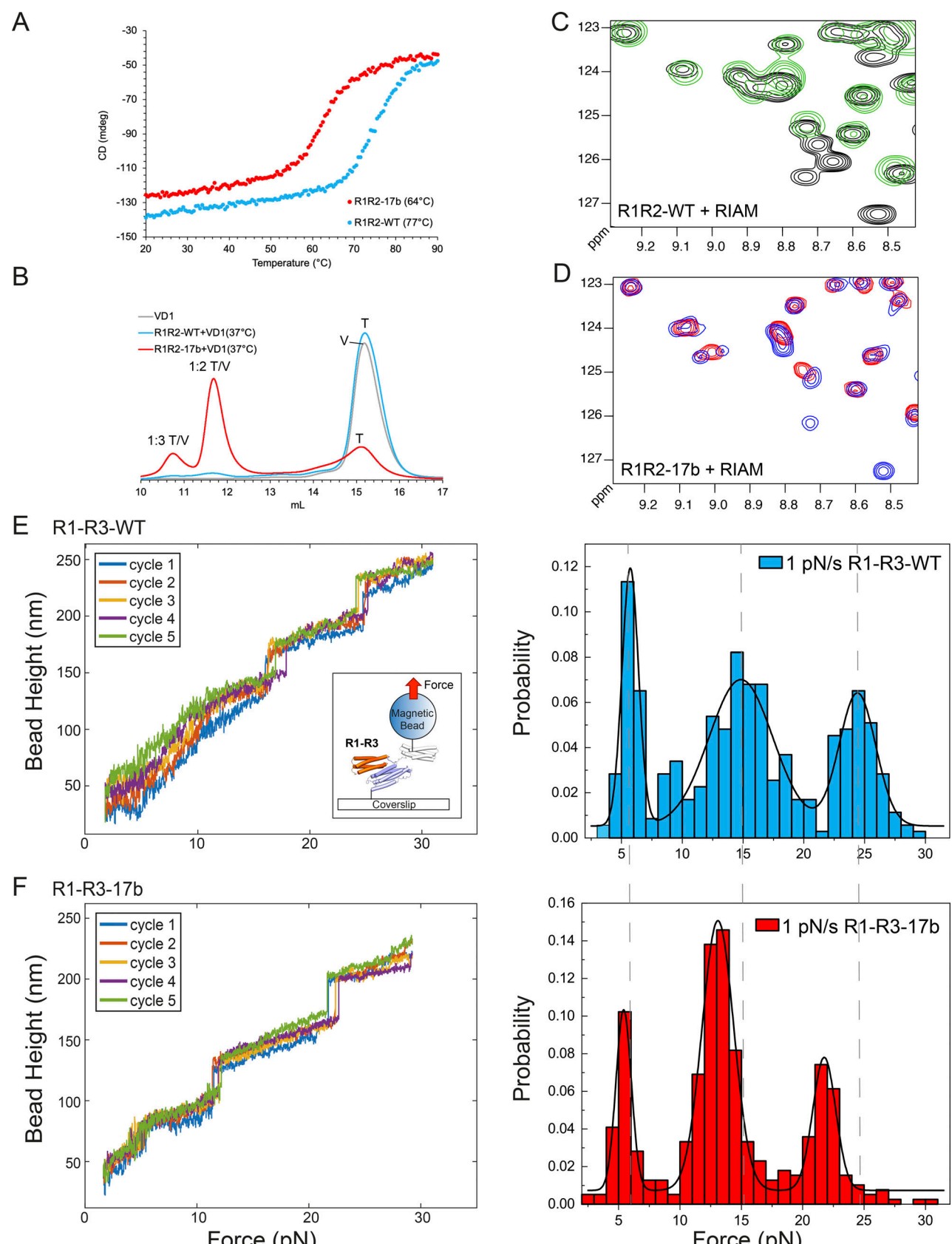

Figure 4. **Exon 17b alters the biophysical, biochemical and mechanical properties of R1R2. (A)** Circular dichroism (CD) analysis using proteins at 16 µM. Denaturation profiles for R1R2-WT (blue) and R1R2-17b (red) were measured by monitoring the change in CD at 222 nm with increasing temperature. Wild-type R1R2 has a melting temperature (T$_m$) of 77°C whereas R1R2-17b unfolds at 64°C. **(B–D)** Characterization of R1R2-WT and R1R2-17b interactions with RIAM and

vinculin. **(B)** Vinculin domain 1 (VD1) binding analyzed by size exclusion chromatography (SEC) using a Superdex-200 gel filtration column. VD1 (V) was incubated at a 3:1 ratio with talin (T), R1R2-WT (blue) or R1R2-17b (red) at 37°C. Minimal binding was observed with the R1R2-WT. However, extensive complexation was observed for R1R2-17b and both 1:2 and 1:3 complex peaks were observed. The calculated molecular weights of the two complex peaks are ∼80 and ∼110 kD, respectively. **(C and D)** RIAM binding to R1R2 analyzed by NMR. $^1$H,$^{15}$N TROSY spectra of 100 μM R1R2-WT (black) and R1R2-17b (blue) alone and upon addition of 300 μM RIAM peptide (green in [C], red in [D]). Extensive chemical shift changes were observed in both confirming RIAM interacts with both versions. **(E and F)** Single-molecule stretching experiments show that 17b inclusion alters the mechanical stability of R1 and R2. The experimental setup is shown in the inset of E. A single molecule of R1-R3-WT (E) and R1-R3-17b (F) is tethered between a paramagnetic microbead and a coverslip. Left panels: Representative time traces of the height of the microbead during repeated force-increase scans at a loading rate of 1 pN s$^{-1}$. Each stepwise bead height increase indicates unfolding of a domain. Five independent force-increase scans are shown by the different color traces. Right panels: Normalized histograms of the unfolding forces of R1-R3-WT (N = 353, seven independent tethers) and those of R1-R3-17b (N = 391, nine independent tethers). The dotted lines indicate the unfolding thresholds for each of the domains in the wild-type R1-R3.

---

of TLN2 (confirmed in Fig. S3 A), whose presence might conceivably mask some effects. Cells expressing similar levels of WT vs. 17b TLN1 (Fig. S3 B) were then plated on fibronectin-coated coverslips for 15 min to 4 h and focal adhesion morphology assessed and quantified. Cells expressing TLN1 17b showed a modest shift toward a greater number of small vs. large adhesions compared to WT TLN1-expressing cells (Fig. 5, A–D). TLN2 knockdown slightly magnified these differences but did not otherwise change the outcomes. No differences in cell spreading were detected between TLN1 WT and 17b (Fig. 5 E). Colocalization with the cell–cell junction marker β-catenin was minimal and similar between isoforms (Fig. S3, C and D); overall cell morphology and F-actin organization were also similar (Fig. S3 E). However, the ratio of vinculin to talin within adhesions was greater for the 17b variant (Fig. 5 B), consistent with our biochemical data (Fig. 4 B), suggesting that the novel TLN1 isoform augments vinculin/talin interaction.

As the shift in adhesion size is suggestive of alterations in cell motility, we measured rate of movement for single cells plated on coverslips. Indeed, cells expressing TLN1 17b migrated ∼2× faster than those expressing WT TLN1 (Fig. 5 F). To determine whether these differences might be related to exchange kinetics of individual molecules within adhesions, we performed fluorescence recovery after photobleaching (FRAP) to assess the turnover. These measurements revealed no difference between TLN1 WT and 17b (Fig. 5, G and H). To assess whether these differences might reflect changes in fibrillar adhesions, we stained cells for tensin-1 as a marker of these structures. No difference in the tensin-1/talin-1 ratio was observed between WT and 17b TLN1 (Fig. 5, I and J).

Given the reduced stability of the R1R2-17b regions (Fig. 4), we also hypothesized that cells might have an altered morphological response to surfaces of different stiffness, which controls force on talin (Kumar et al., 2016). Stiffness sensing often examines spread cell area, which increases with increasing stiffness. However, when WT and 17b talin-1-expressing cells were plated on fibronectin-coated polyacrylamide gels of variable stiffness, no difference in cell spread area was detected (Fig. S3 F).

Together, these cellular experiments support a role for TLN1 exon 17b in altering focal adhesion formation, but surprisingly, this altered adhesion did not lead to changes in cellular substrate sensing, at least in the conditions tested. It is possible that the effects of this variant are less evident in baseline conditions i.e., without an additional stimulus. Nevertheless, cells expressing the TLN1 exon 17b isoform showed increased motility, indicating a functional impact of TLN1 alternative splicing on cell migration.

## Conclusions

In summary, we reported the discovery of a novel cassette exon 17b in the *TLN1* gene, which alters the talin-1 protein coding sequence. This cassette exon can already be detected in healthy tissue but is highly enriched in several subtypes of cancer. Moreover, splicing of TLN1 exon 17b can be controlled by TGF-β signaling pathways.

Until now, TLN1 has been thought to have a single isoform, so this discovery of an alternatively spliced isoform has significant implications for our understanding of integrin signaling. TLN1 forms a mechanosensitive signaling hub on the inside of the integrin adhesion complexes (Goult et al., 2021) where the 13 force-dependent binary switches in talin open and close in response to mechanical signals.

Whilst the full functional significance of talin-1 alternative splicing is still to be determined, we showed that the inclusion of this extra 17 amino acids markedly alters the stability of the R1 and R2 switches in the talin rod such that less force is required to open them. FRET-based tension sensors found that the average tension on talin in cells is ∼5 pN (Kumar et al., 2016) and relatively uniform across the cell. However, the use of sensors that report tension above 7–11 pN revealed that a fraction of talin experiences tension above this range (Austen et al., 2015) and threshold tension sensors for integrins detected spikes >20 pN (Wang and Ha, 2013). We suggest that these transient high tension events trigger domain unfolding, which can be maintained at the lower average tension. This point was also evident in Margadant et al., (2011) where measurement of the end-to-end distance of talin in cells revealed cycles of stretch and relaxation, with changes in length between 50–350 nm. When we simulated responses to force based on these end-to-end fluctuations, we saw that average force per talin at <10 pN but with spikes up to ∼20 pN can reproduce these changes (Yao et al., 2016).

Modification of the mechanical stability of talin rod domains has been shown to alter the global mechanotransduction of the cell (Elosegui-Artola et al., 2014; Haining et al., 2018). The consequence of this alteration in the R1 and R2 switches is that the binary patterns of information written into the shape of each talin molecule (Goult, 2021) will be different dependent on the status of the 17b exon, as different signaling outputs will be

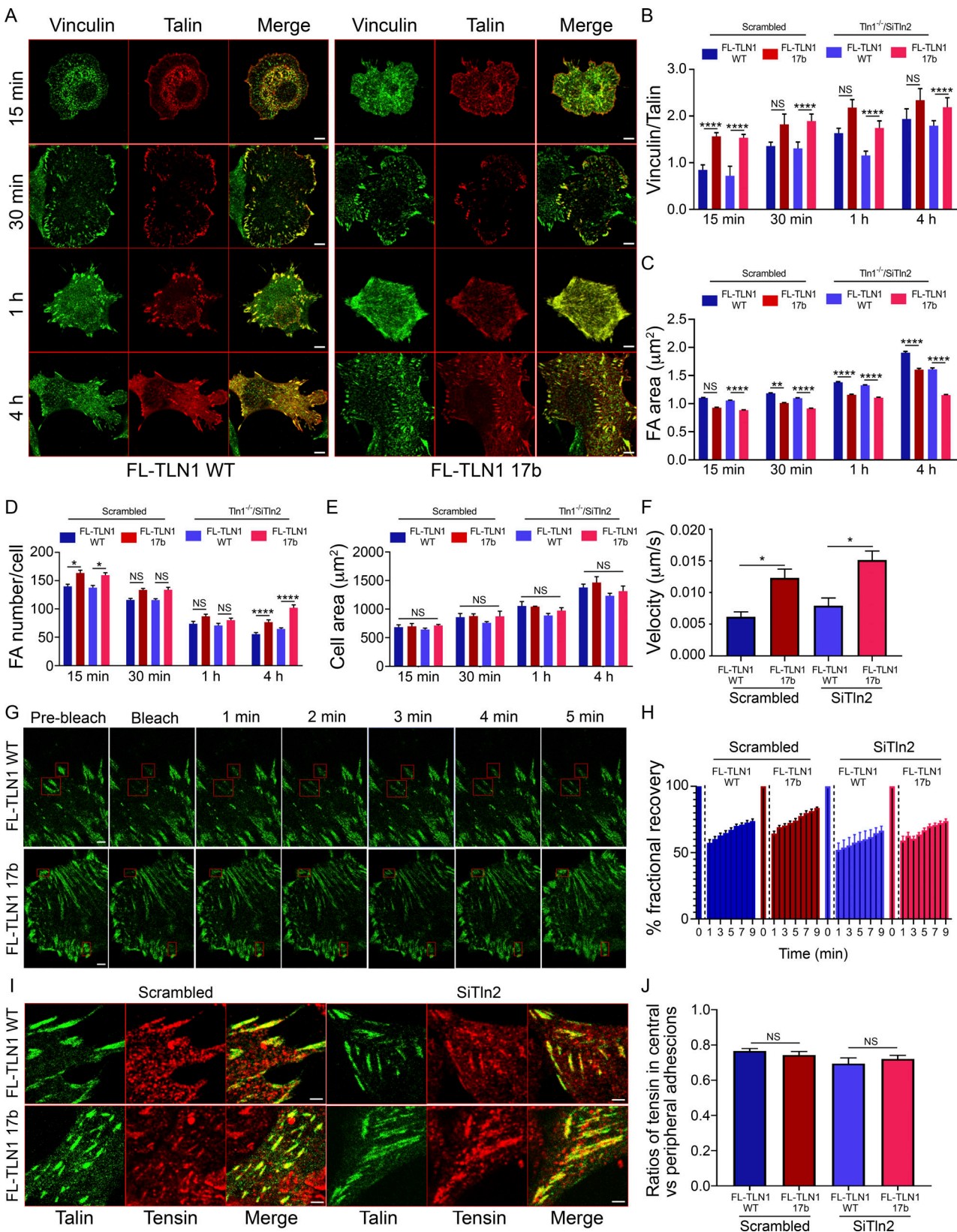

Figure 5. **The TLN1 exon 17b isoform alters focal adhesions in cells. (A)** Representative images of focal adhesions in Tln1$^{-/-}$/Tln2$^{Si}$ MEFs expressing full-length talin-1 WT and exon 17b splice variant co-stained with vinculin at different plating times. Scale bar, 5 μm. **(B)** Quantification of vinculin to talin-1 ratio in A. Data are mean ± SEM, $N$ = 290–1,987 adhesion. **(C)** Quantification of FA area in A. Data are mean ± SEM. of 3,196-4,658 adhesions. **(D)** Quantification of focal adhesion number in A. Data are mean ± SEM of 25–60 cells. **(E)** Quantification of cell area. Data are mean ± SEM. $N$ = 3. **(F)** Migration velocity of cells expressing either full-length talin-1 WT or exon 17b splice variant. Data are mean ± SEM. $N$ = 10 cells. **(G)** Representative images of FRAP experiment in Tln1$^{-/-}$/

Tln2[Si] MEFs expressing full-length talin-1 WT and exon 17b splice variants. Boxed regions represent bleached FAs. Scale bar, 1 µm. **(H)** Fractional recovery percentage of GFP-talin-1 within individual focal adhesions was plotted against time. Data are mean ± SEM. N = 10 cells/sample. **(I)** Representative images of focal adhesions in Tln1[−/−]/Tln2[Si] MEFs expressing full-length talin-1 WT and exon 17b splice variant co-stained with tensin-1. Scale bar, 5 µm. **(J)** Quantification of ratio of tensin-1 mean fluorescence within central and peripheral talin adhesions of a cell. Data are mean ± SEM. N = 52–84 cells. Statistical significance was calculated by one-way analysis of variance (ANOVA) for each time point (B–E) between scrambled and knockdown conditions (B–E, F, H, and J). *, P < 0.05; **, P < 0.01; ****, P < 0.0001.

generated in response to the same amount of mechanical signals. Our recent analysis on the scale of talin domains unfolding (Barnett and Goult, 2022) revealed that helical bundle unfolding introduces a large extension into the length of the talin molecule, and R1R2 unfolding would introduce ~115 nm increase in length, relocating domains R3-R13 and all attached ligands further away from the membrane altering the spatial organization of molecules.

Exon 17b inclusion results in more small and fewer large adhesions, and with a greater vinculin/talin ratio within the adhesions, perhaps reflecting the decreased mechanostability of the R1-R2 regions. Cells expressing the 17b isoform also migrated faster, consistent with the shift toward smaller adhesions. It is tempting to speculate that the decreased stability of R1R2 and the concurrent increased vinculin binding to this region, results in altered adhesion dynamics by altering how force and cytoskeletal connections load onto the talin molecule maintaining adhesions in a more nascent state. It is also possible that the 17 amino acids, which extend out of both the open (Fig. 3 A) and closed (Fig. 3 G) states of TLN1 present a novel binding motif for some, as of yet, unidentified ligand. Or it might be that the insertion, simply by altering the stability of the R1R2 domains, impacts the mechanical response of talin, and this is sufficient to exert global changes in cell signaling.

There now exist vast amounts of publicly available RNA-Seq datasets from researchers looking at many different systems and conditions, and this provides a wealth of information on this altered splicing event of TLN1. Our DJExpress pipeline enabled us to establish when and where the exon is included and that its inclusion can be regulated by TGF-β/Smad3 in certain cellular settings. TGF-β signaling is involved in many processes related to cancer malignancy and its use as a therapeutic target is actively investigated in the clinics (Hanahan and Weinberg, 2011; Derynck et al., 2021). The enrichment of TLN1 exon 17b in several cancer subtypes together with the role of TGF-β signaling in its regulation thus could have far-reaching implications in the field of cancer biology.

Interestingly, we provided evidence that the *TLN1* splice variant is part of a concerted splicing program as we see inverse reciprocity between *TLN1* exon 17b inclusion and *CLSTN1* exon 11 exclusion. Further work would be required to fully appreciate the interplay between these previously unconnected proteins, but we speculate that they might form part of a change in the cellular program linked to the epithelial-mesenchymal transition (EMT).

We showed previously how post-translation modification of TLN1 via cyclin-dependent kinase 1 (CDK1) can alter the mechanical response of the R7R8 region of talin, thereby changing the switch patterns that form (Gough et al., 2021). Here we show that alterations in the primary sequence of TLN1 via alternative splicing can also impact on the switch patterns. This finding

provides additional support to the emerging idea that mechanical signaling through TLN1 can be augmented via numerous chemical modifications providing a way to integrate mechanical and biochemical signals into signaling outputs. It will be important in future work to evaluate the status of the TLN1 isoforms being expressed in that cell type as alterations in cell signaling can alter mechanotransduction via modifying talin at the protein isoform level.

## Materials and methods
### Human tumor and cell line datasets
RNA-Seq data alignment, splice junction read quantification, and sample quality-based filtering using 9,842 TCGA (Tomczak et al., 2015) tumor tissue samples across 32 different tumor types, 3,235 normal post-mortem tissue samples from GTEx (Lonsdale et al., 2013) and 1,019 cancer cell lines from the DepMap project (release 21Q3, www.depmap.org) was previously done as previously described (Kahles et al., 2018; Gallego-Paez and Mauer, 2022). Clinical data associated with TCGA tumor samples was obtained using *TCGAbiolinks* package (Colaprico et al., 2016). Drug treatment and CRISPR screens data from cancer cell lines was retrieved from DepMap project portal (release 21Q3, www.depmap.org). RNA-Seq data from TGF-β and EGF-treated HeLa cells was downloaded from the GEO database (accession number GSE72419) and raw data were processed with *DJExpress* (Gallego-Paez and Mauer, 2022).

### Differential junction expression (DJE) analysis
Differential junction expression analysis was carried out using *DJExpress* tool (Gallego-Paez and Mauer, 2022). Briefly, quantified splice junction counts are gene-annotated, filtered by expression threshold (10 minimum read count mean per junction), and transformed to $\log_2$-counts per million (logCPM). Observation-level weights for each junction were computed using *limma* method (Ritchie et al., 2015) and linear modeling with empirical Bayes moderated *t*-statistics were implemented using *DJEanalyze* function to measure the significance level of observed changes in junction expression and usage between GTEx normal tissue and TCGA tumor tissue samples, between Fibroblasts "healthy" control cells and CCLE cancer cell lines and between TGF-β/EGF-treated and untreated HeLa cells. Genewise splice graphs and gene model plots with exon-to-protein domain annotation are produced using *DJEplotSplice* function.

### Associations between TLN1 junction expression and functional genomics data
Significant linkages between *TLN1* exon 17b-related junction expression and drug treatment response or gene dependency

values in cancer cell lines were identified using correlation matrix operations implemented as part of the *DJEvsTrait* function from *DJExpress* (test.type = "Correlation", cor.method = "bicor", P value = 0.05). *SpliceRadar* function was used for radar chart representation of top-ranked significant correlation coefficients. The clusterProfiler package version 4.4.2 (Yu et al., 2012) was used to identify enriched KEGG pathways within correlated gene dependencies to TLN1 junction expression, ranked by absolute correlation coefficient (adjusted P value <0.05 for significantly enriched pathways).

### Reverse transcription PCR (RT-PCR)

Total RNA was isolated from cancer cell lines and 1 µg was converted into cDNA using ProtoScript II First Strand cDNA Synthesis Kit (NEB). RT-PCR was performed with primers spanning the region between TLN1 exon 17 and exon 18 (TLN1_17b_fwd 5′-CAAGCAGCTGGGAACGTG-3′, TLN1_17b_rev 5′-CTGAAGTCCCGAGTCCTCTG-3′) and the region between CLSTN1 exon 10 and exon 12 (CLSTN1_11_fwd 5′-GACTCTCTATGTGGATGGCACG-3′, CLSTN1_11_rev 5′-CCTTGCAGGTATACAGACAGTCG-3′). The RT-PCR reactions were carried out at 94°C for 30 s, 55°C for 60 s, and 68°C for 30 s for 30 cycles. The resulting PCR products were separated on a 1% agarose gel at 100 V for 1 h.

### Cell line and organoid culture

BT20, BT549, MCF7, MCF10a, MDA-MB231, MDA-MB453, SKBR3, SUM44PE, T47D, and ZR751 cell lines were obtained from the American Type Culture Collection (ATCC), STR type verified by PCR, and cultured as described previously. Derivation, establishment and culturing for the PDO models P008, P008::ΔCDH1 and KCL320 is extensively described in the supplementary materials of Rätze et al. (2022). Culturing of PDO models HUB-72T (breast) and HUB-93T (breast) was published previously and are distributed by HUB ORGANOIDS (Sachs et al., 2018; Driehuis et al., 2019). Generation of E-cadherin knock-out MCF7::ΔCDH1 and MCF10a::ΔCDH1 using lentiviral CRISPR-Cas9 CDH1 targeted editing has been described previously in (Hornsveld et al., 2016). All cell lines and organoid models were routinely tested for Mycoplasma infection.

### Protein expression and purification

Human talin-1 R1R2-WT (residues 482–786) and R1R2-17b (residues 482–786 + additional 17 residues) were produced as codon optimized synthetic genes in pET151 plasmids (GeneArt). The vinculin Vd1 construct was also in pET151 as described previously (Wang et al., 2021). The RIAM peptide (residues 4–30; sequence SEDIDQMFSTLLGEMDLLTQSLGVDTC) was synthesized by GLBiochem. For the magnetic tweezers experiments human talin-1 R1-R3-WT (residues 482–911) and R1-R3-17b (residues 482–911 + additional 17 residues) were cloned into a pGEX vector with Halo and Avi-tags as described in Yao et al. (2016).

All constructs were transformed into BL21(DE3)* *E. coli* cells and grown in lysogeny broth + 100 µg/ml ampicillin at 37°C. Once $OD_{600}$ had reached 0.7–0.8, the cultures were induced with 0.1 mM IPTG and incubated overnight at 20°C. Following harvesting, the expressed proteins for biochemical analysis were purified using Nickel-NTA affinity chromatography. In short, after harvesting, the cell pellets were resuspended in 20 mM Tris-HCl, pH 8, 500 mM NaCl, and 50 mM imidazole and lysed using sonication. Sonicated cell lysate was loaded into a 5 ml HisTrap HP column (GE Healthcare) and purified by nickel affinity chromatography. The eluted protein was dialyzed overnight into 20 mM Tris-HCl, pH 8, 50 mM NaCl, and TEV protease (Invitrogen) was added to remove the His-tag. Proteins were further purified using a HiTrap Q HP cation exchange column (GE Healthcare). Further details are available in Khan et al. (2021). The GST-tagged constructs for the single-molecule studies were purified using glutathione Sepharose (GE Healthcare) and eluted by TEV cleavage.

The recombinant protein expression constructs have been deposited in Addgene at http://www.addgene.org/ben_goult.

### AlphaFold structural modelling

To produce the structural models of human R1R2-WT and R1R2-17b, the 3D protein structure prediction tool, AlphaFold (Senior et al., 2020) was used. For this the sequence R1R2 +/−17b was submitted to the software and the structural models visualized in PyMOL and compared against the crystal structure of mouse R1R2 (Protein Data Bank accession no. 1SJ8 [Papagrigoriou et al., 2004]).

### Circular dichroism (CD)

Circular dichroism was performed using a JASCO J-715 spectropolarimeter with 0.5 mg/ml protein samples. The far-UV spectra were collected between wavelengths of 200–260 nm, at six scans at a speed of 50 nm/min, 1 nm step resolution and a band width of 1 nm. CD melting curve data were collected at a wavelength of 222 nm, between 20–90°C, 1°C step resolution, and 1 nm band width.

### Size exclusion chromatography (SEC)

SEC analysis of R1R2-WT, R1R2-17b and vinculin domain 1 (VD1) was done at room temperature. The total volume of samples was 100 µl, with protein concentrations between 100–400 µM (experiment dependent). The samples were loaded and ran using a Superdex 200 Increase 10/300 GL column. Preincubation at 37°C was done for 1 h immediately prior to loading.

### Nuclear magnetic resonance (NMR)

NMR samples were prepared in 50 mM NaCl, 15 mM $NaH_2PO_4$, 6 mM $Na_2HPO_4$, 2 mM DTT, pH 6.5, 5% (vol/vol) $D_2O$. All experiments were run at 298K on a Bruker Avance III 600 MHz NMR spectrometer equipped with CryoProbe. Data were processed with Topspin and analyzed using CCPN Analysis (Skinner et al., 2015).

### Single-molecule manipulation experiments

In the magnetic tweezer experiments, forces are applied to a protein of interest via superparamagnetic microbead and a 576 bp DNA linker using a custom magnetic tweezers platform that can exert forces up to 100 pN with ~1 nm extension resolution for tethered bead at 200 Hz sampling rate (Chen et al., 2011). The C-terminal of the protein of interest (R1-R3-17b or R1-R3-WT) was tethered to a Halo-ligand-coated coverslip via the

C-terminal Halo-tag. The biotinylated N-terminal Avi-tag is linked to the biotinylated end of a 576 bp DNA linker via a traptavidin (Chivers et al., 2010). The other end of the DNA linker, which is labeled with a thiol group, is covalently attached to a superparamagnetic microbead (Dynabeads M270-epoxy). This way, a molecular tether consisting of the protein of interest and a DNA linker is spanned between the coverslip glass surface and the superparamagnetic microbead. Further details can be found in our previous publication (Yao et al., 2016). All unfolding experiments were carried out in PBS, 3% BSA, 1 mM dithiothreitol and 0.1% Tween-20.

For given magnets and bead, the force is solely dependent on the magnet-bead distance $F(d)$, which can be calibrated based on a method described in our previous publication, which has an ~10% uncertainty due to the heterogeneous bead sizes (Chen et al., 2011). The force loading rate control is achieved by decreasing the magnet-bead distance $d(t)$ in a manner such that the force increases at a constant rate $r$ (Zhao et al., 2017).

### Mammalian expression plasmids, knockdown, and immunofluorescence

Full-length WT talin-1 plasmid containing internal GFP and RFP (Kumar et al., 2016) was used in this study. Nucleotides corresponding to the 17-residue region of the 17b splice variant were cloned into WT talin-1 by Gibson assembly. Briefly, the WT talin-1 plasmid was digested with NotI and XhoI. Primers containing overhangs of nucleotides corresponding to the 17-residue region of the splice variant were used to amplify R2. Two fragments containing nucleotides between the NotI site to R1 domain and beyond R2 domain to XhoI site were also amplified by PCR. All fragments were incubated with Gibson assembly mix (New England Biolabs) as per manufacturer's instructions.

Talin-2 was knocked down in Tln1$^{-/-}$ cells (Priddle et al., 1998) by transient transfection of Tln2 siRNA (ONTARGET-plus Smartpool siRNA, Catalog ID:L-065877-00-0005, Horizon Discovery) and scrambled siRNA (AM4636; Ambion), using Lipofectamine RNAimax, as described previously (Kumar et al., 2016). Knockdown was confirmed by immunoblotting for talin-2 using mouse anti-talin-2 antibody (AC14-0126; Abcore). These cells were transfected with full-length WT talin-1 and full-length talin-1-17b variant, using Lipofectamine 2000 (Thermo Fisher Scientific) for 24 h and trypsinized for all experiments at this time point. For early adhesion analysis, Tln1−/−transfected cells were held in suspension for 30 min and plated on fibronectin-coated glass bottom dishes for 15 min, 30 min, 1 h, and 4 h. Cells were fixed in 4% paraformaldehyde and counterstained with Alexa 405 phalloidin (Thermo Fisher Scientific) and anti-vinculin antibody (V9131; Sigma-Aldrich). Cells were imaged using a 63× objective on a Leica SP8 confocal microscope. ImageJ was used to assess cell area, focal adhesion area, focal adhesion number and mean fluorescence intensities of vinculin and talin within each adhesion. To assess substrate stiffness–dependent cell spreading, cells were trypsinized and plated on either fibronectin-coated (10 μg/ml) glass-bottom dishes or polyacrylamide gels of varying stiffness. At 6 h, cells were fixed in 4% paraformaldehyde and counterstained with Alexa 647 phalloidin (Thermo Fisher Scientific). The cell area was calculated using ImageJ.

For tensin-1 localization, cells were plated for 48 h and counter-stained with tensin-1 (SAB4200283; Sigma-Aldrich). Talin adhesions that were within 7 μm from centroid of the cell were considered central adhesions and tensin intensity was quantified within these adhesions. Mean of tensin intensity within central adhesions was divided by mean of tensin intensity of peripheral adhesions in each cell and the ratio was plotted. β-catenin staining was also performed on densely seeded cells using anti-β-catenin antibody (9562; CST).

### Live-cell imaging

MEFs expressing full-length WT talin1 and full-length talin1-17b variant expressing cells were seeded on glass-bottom dishes for 8 h. Hoechst (1 μg/ml) was added for 1 h before imaging. Live-cell imaging was performed at 20X on Leica SP8 live-imaging system. Images were acquired every 5 min for 4 h. The movement of the cell was tracked in each frame in ImageJ using the Track-mate plugin and the velocity of each cell was calculated.

FRAP was performed to assess the dynamics of focal adhesions containing WT talin1 or 17b splice variant. Live-cell imaging was performed at 63X on Leica SP8 live-imaging system. Before photobleaching, three pre-bleach images of GFP-talin were acquired, using a 488 nm laser set at 10% of the maximum power. Photobleaching of GFP was conducted for 10 s at 100% laser power. Fluorescence recovery images were acquired every 30 s for 9 min using a 488 nm laser set at 10% of the maximum power. The mean fluorescence intensity pre-bleach was set to 100%. Photobleaching due to continuous illumination during recording was corrected by normalizing the fluorescence intensity at the FA with total cell fluorescence intensity. Corrected recovery fluorescence intensities were normalized to the pre-bleach intensity. The intensity was considered 100 and 0% for pre-bleach and bleach points. The fractional recovery post-bleach was calculated by normalizing the corrected recovery fluorescence intensities at each time point to pre-bleach intensity.

### Online supplemental material

Fig. S1 shows the *TLN1* gene model and its alignment with other primates, a summary of *TLN1* differentially expressed junctions in tumor vs. normal tissue, the RT-PCR analysis of the *TLN1* exon 17b and *CLSTN1* exon 11 in 2D cell lines and 3D organoid lines, the AlphaFold model of R1R2-17b, and the SEC profiles of R1R2-WT and R1R2-17b. Fig. S2 shows the inverse correlation between *CLSTN1* exon 11 and *TLN1* exon 17b expression data in cancer patients and cell lines. Fig. S3 shows the additional cell biological experiments and controls that accompany Fig. 5.

## Acknowledgments

We thank Sridevi Jaksani for technical support with organoid culturing.

B.T. Goult was funded by Biotechnology and Biological Sciences Research Council (grant BB/S007245/1) and Cancer Research UK Program (grant DRCRPG-May21\100002). J. Mauer and L.M. Gallego-Paez were funded by Merck KGaA, Darmstadt, Germany (CrossRef Funder ID: 10.13039/100009945). M.A. Schwartz was funded by United States Public Health Service

(grant R01 GM047214-25). J. Yan was funded by The Ministry of Education under the Research Centres of Excellence programme and Singapore Ministry of Education Academic Research Funds Tier 2 (MOE-T2EP50220-0015). T. Koorman and P.W.B. Derksen received financial support from Breast Cancer Now (2018novPCC1297) which is supported by Pfizer, the European Union's FET Proactive program under the grant agreement no. 731957 (MECHANO-CONTROL), and COST action LOBSTERPOT (CA19138), supported by COST (European Cooperation in Science and Technology).

Author contributions: L.M. Gallego-Paez performed the bioinformatic analyses with support from C.-Y. Lee, T. Koorman, C-Y. Lee, and N. Grexa performed the RT-PCR analyses. W.J.S. Edwards. performed the biochemical and biophysical analysis and the computational modelling. Y. Guo and J. Yan performed the single-molecule analysis. M. Chanduri performed the cell biology and stiffness sensing experiments. T. Koorman. and P.W.B. Derksen provided conceptual input. J. Mauer and B.T. Goult conceived the study, supervised the project and wrote the paper with input from all authors. M.A. Schwartz supervised the cell biology experiments. All authors have read the manuscript and were given the opportunity to provide input.

Disclosures: All authors have completed and submitted the ICMJE Form for Disclosure of Potential Conflicts of Interest. L. Gallego-Paez reported "At the time this study was conceived, Dr. Gallego-Paez was an employee of BioMed X Institute (GmbH, Heidelberg, Germany), and her research was funded by the healthcare business of Merck KGaA (Darmstadt, Germany; CrossRef Funder ID: 10.13039/100009945). The healthcare business of Merck KGaA had no part in the study design and data collection, analysis, or interpretation of the results but provided feedback regarding the general research strategy". J. Mauer reported "At the time this study was conceived, Dr. Mauer was an employee of BioMed X Institute (GmbH, Heidelberg, Germany), and his research was funded by the healthcare business of Merck KGaA (Darmstadt, Germany; CrossRef Funder ID: 10.13039/100009945). The healthcare business of Merck KGaA had no part in the study design and data collection, analysis or interpretation of the results but provided feedback regarding the general research strategy. Dr. Mauer is currently an employee at Novartis Pharma AG (Basel, Switzerland). Novartis Pharma AG had no part in this study". No other disclosures were reported.

Submitted: 6 September 2022

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

# Supplemental material

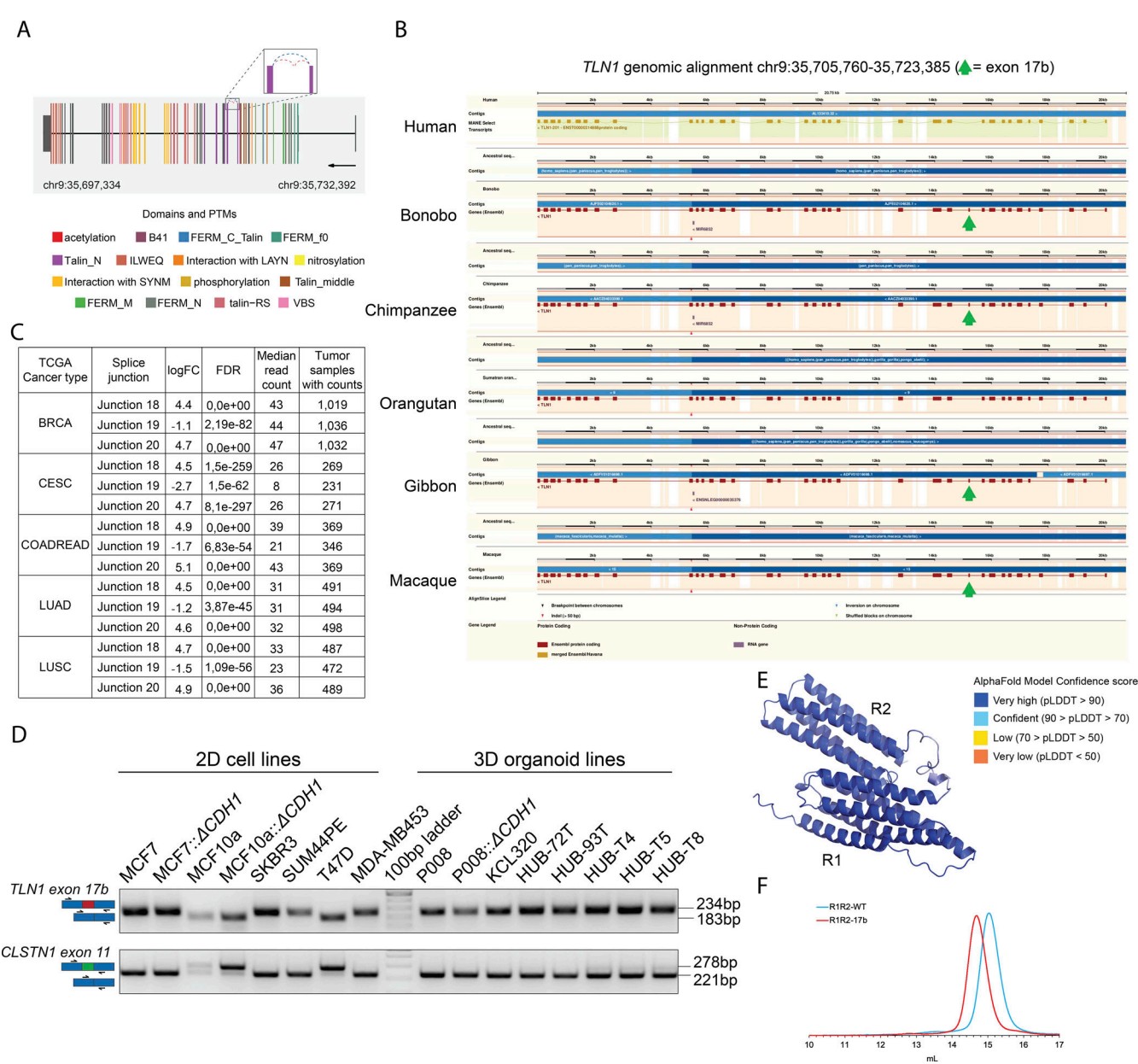

Figure S1.    **Supporting analysis of Exon 17b. (A)** *TLN1* gene model plot with exon-to-protein domain annotation and differentially spliced junctions marked as dashed arcs connecting upstream and downstream exons. Coding and untranslated region (UTR) exons are illustrated as long and short exons, respectively. Colors within exonic regions indicate the presence of protein domains and/or post translational modifications (PTMs) annotated within the Prot2HG protein domain database (Stanek et al., 2020). The differentially expressed junctions in *TLN1* are located within the TLN1 coding region. **(B)** Alignment of *TLN1* gene structure in humans and primates. The green arrows indicate the position of *TLN1* exon 17b, which is annotated in bonobo, chimpanzee, gibbon, and macaque but not documented in humans and orangutan. **(C)** Table showing a summary of *TLN1* differentially expressed junctions (tumor vs. normal tissue) in five example tumor types from TCGA (breast invasive carcinoma [BRCA], cervical squamous cell carcinoma [CESC], colon and rectal adenocarcinoma [COADREAD], lung adenocarcinoma [LUAD], and lung squamous cell carcinoma [LUSC]; LogFC = log fold-change). **(D)** *TLN1* exon 17b and *CLSTN1* exon 11 inclusion/exclusion in cancer cell lines grown in 2D and 3D. Top: *TLN1* exon 17b expression status in 2D breast cancer cell lines and 3D breast cancer organoid models was assessed using RT-PCR. RT-PCR was performed with primers flanking exon 17b (primer positions indicated by black arrows). Exon 17b spans 51 base pairs (bp) and exon 17b inclusion results in an amplicon size increase from 183 bp to 234 bp analyzed on an agarose gel. Bottom: The *CLSTN1* exon 11 expression status in 2D breast cancer cell lines and 3D breast cancer organoid models. RT-PCR was performed with primers flanking exon 11 (primer positions indicated by black arrows). Exon 11 skipping results in an amplicon size reduction from 278 bp to 221 base pairs. In all cell lines tested the inverse splicing pattern between *TLN1* exon 17b and *CLSTN1* exon 11 is observed. Note that the organoid models grown in 3D all express the 17b inclusion version of talin-1 and the exon 11 exclusion version of calsyntenin-1. **(E)** AlphaFold model of the R1R2-17b structure colored by Model Confidence score. pLDDT = predicted local distance difference test score (0–100). This is a per-residue confidence score, with values greater than 90 indicating high confidence, and values below 50 indicating low confidence. All residues in the structural model have a score >70. **(F)** Size exclusion chromatography (SEC) using a Superdex-200 gel filtration column of talin R1R2-WT (blue) or R1R2-17b (red). The calculated molecular weights of the two monomer peaks are ~31 and ~33 kD, respectively.

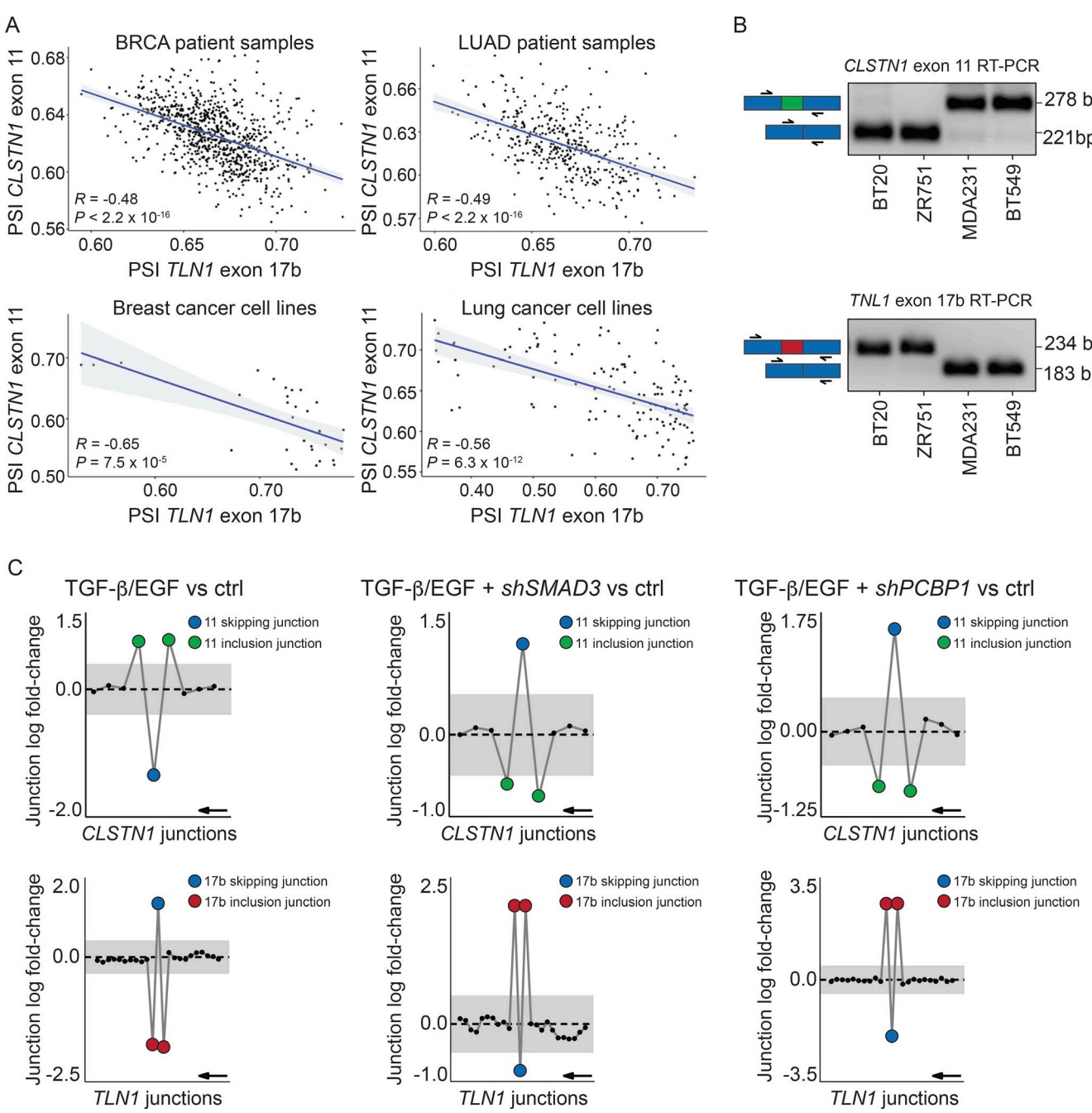

Figure S2.   **CLSTN1 exon 11 expression is inversely correlated with *TLN1* exon 17b expression in cancer patients and cell lines. (A)** Correlation analysis of percent spliced-in (PSI) values of *TLN1* exon 17b and *CLSTN1* exon 11 in tumor patient tissue and cancer cell lines. *TLN1* exon 17b and *CLSTN1* exon 11 splicing is inversely correlated, suggesting nearly trans-mutually exclusive splicing regulation of these two events (*R* = Pearson correlation coefficient, P = adjusted P value). **(B)** RT-PCR validation of *CLSTN1* exon 11 expression in four representative breast cancer cell lines. RT-PCR was performed with primers flanking exon 11 (primer positions indicated by black arrows). Exon 11 skipping results in an amplicon size reduction from 278 bp to 221 base pairs. BT20 and ZR751 cell lines show exon 11 skipping, whereas MDA231 and BT549 cell lines show exon 11 inclusion. The RT-PCR analysis of TLN1 exon 17b expression in cancer cell lines shown in Fig. 2 B was repurposed here to illustrate the inverse splicing pattern with *CLSTN1* exon 11. **(C)** Analogous analysis to Fig. 2 E that reveals dynamic *CLSTN1* exon 11 splicing in response to combined TGF-β/EGF treatment. Gene-wise splice plots of *CLSTN1* junction expression in HeLa cells, which show baseline skipping of exon 11. The analysis of the *TLN1* exon 17b expression in HeLa cells shown in Fig. 2 E was repurposed here to illustrate the inverse splicing pattern with *CLSTN1* exon 11. (The plots shown in this figure were generated by *DJExpress*-based re-analysis of RNA-Seq data from GSE72419; gray area indicates the log-fold change cut-off (|logFC| > 0.5). Inclusion junctions are shown in red, skipping junctions are shown in blue. Junctions with FDR > 0.05 for absolute or relative logFC (or both) are shown in black. Black arrow indicates the direction of transcription on the reverse strand).

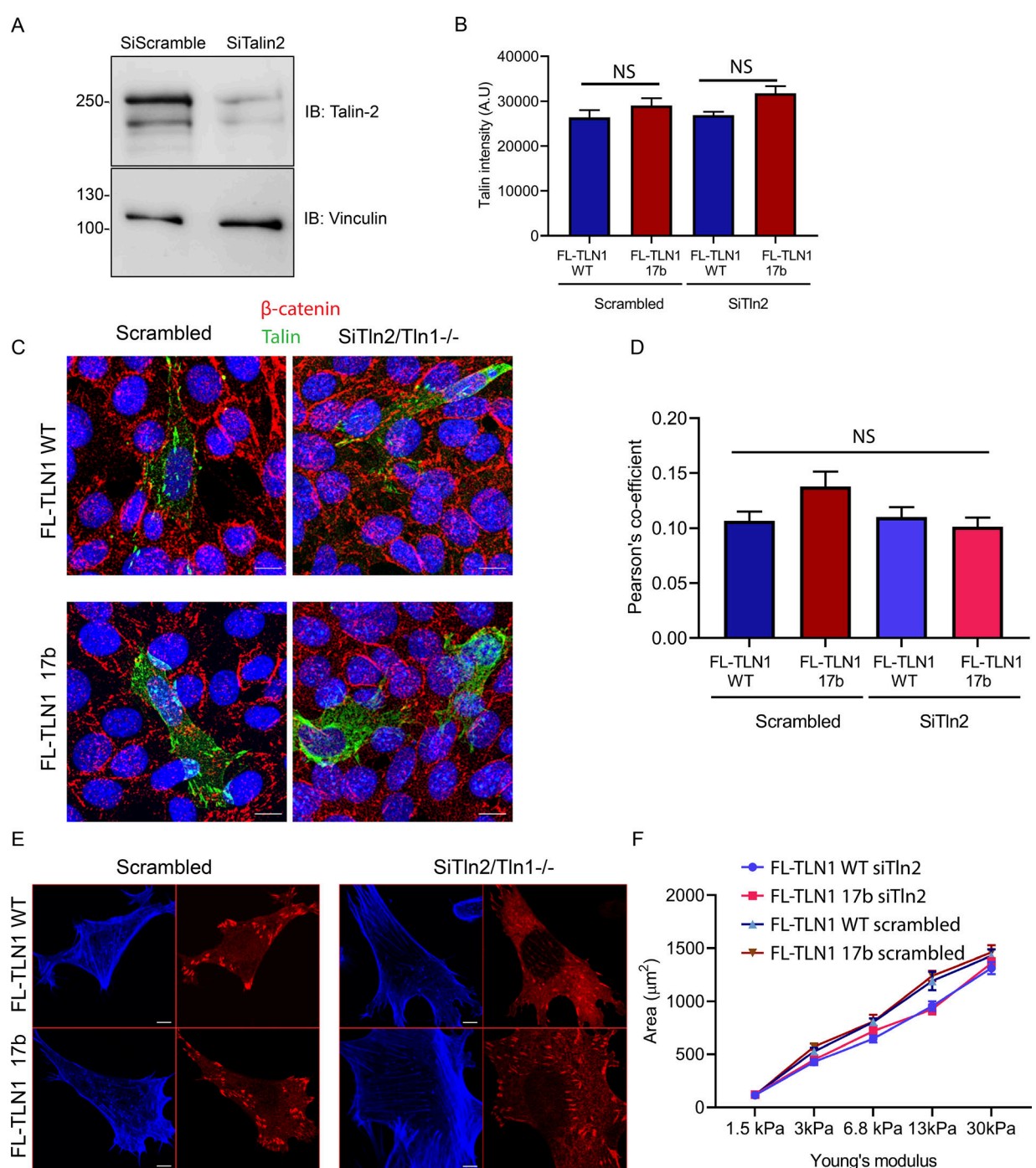

Figure S3. **Supporting cell biological analysis of Talin exon 17b isoform. (A)** Immunoblots of talin-2 in Tln1$^{-/-}$ cells after transfection with scrambled siRNA or talin-2 siRNA. Vinculin was used as a loading control. **(B)** Quantification of talin fluorescence intensity in MEFs expressing full-length talin-1 WT and exon 17b splice variant. Data are mean ± SEM. $N$ = 451–2,018 cells. **(C)** Representative images of cell junctions in Tln1$^{-/-}$/Tln2$^{Si}$ MEFs expressing full-length talin-1 WT and exon 17b splice variant co-stained with β-catenin. Scale bar, 5 μm. **(D)** Quantification Pearson's coefficient of co-localization of β-catenin with talin. Data are mean ± SEM. $N$ = 75–100 cells. Statistical significance was calculated by one-way analysis of variance (ANOVA) for each time point (B and D) between scrambled and knockdown conditions. **(E)** Representative images of Alexa 405 phalloidin staining (blue) to visualize actin in Tln1$^{-/-}$/Tln2$^{Si}$ MEFs expressing full-length talin-1 WT and exon 17b splice variant (red). Scale bar, 5 μm. **(F)** Cell area of Tln1$^{-/-}$/Tln2$^{Si}$ expressing either WT or 17b splice variant form of talin-1, plated for 6 h on polyacrylamide gels of varying stiffness. Data are mean ± SEM. $N$ = 3.

