## [Peer Review File · The Journal of Cell Biology]

TLN1 contains a cancer-associated cassette exon that alters talin-1 mechanosensitivity

Lina Gallego-Paez, William Edwards, Manasa Chanduri, Yanyu Guo, Thijs Koorman, Chieh-Yu Lee, Nina Grexa, Patrick Derksen, Yan Jie, Martin Schwartz, Jan Mauer, and Benjamin Goult

Corresponding Author(s): Benjamin Goult, University of Kent

Review Timeline:

Submission Date:	2022-09-06
Editorial Decision:	2022-09-28
Revision Received:	2023-01-08
Editorial Decision:	2023-02-10
Revision Received:	2023-02-16

Monitoring Editor: Johanna Ivaska

Scientific Editor: Tim Fessenden

Transaction Report:

DOI: <https://doi.org/10.1083/jcb.202209010>

September 28, 2022

Re: JCB manuscript #202209010

Dr. Benjamin Thomas Goult
University of Kent
School of Biosciences,
I4/24 Ingram Building
Canterbury, Kent CT2 7NJ
United Kingdom

Dear Dr. Goult,

Thank you for submitting your manuscript entitled "A novel cancer-associated cassette exon in TLN1 alters Talin 1 mechanosensitivity". The manuscript was assessed by expert reviewers, whose comments are appended to this letter.

In the reviews from two well-established experts in field, there were some significant concerns that need to be addressed. However, our overall conclusion is that this study could be potentially appropriate for publication in JCB if appropriately revised. Consequently, we invite you to submit a revision if you can address the reviewers' concerns.

As you will find, some of the concerns of Rev1 and Rev2 are overlapping. Both highlight the need to refine the effects of the splice variant on focal adhesions. In addition, Rev2 makes an interesting suggestion of analyzing cells seeded onto micropatterned substrates. While we find this interesting we leave it to you to evaluate whether to carry out these studies. Rev2 also raises an interesting point on the forces to which talin is subjected, which we believe you can address in the text.

We strongly encourage you to make every effort to resolve the concerns of these conscientious reviewers. While you are revising your manuscript, please also attend to the following editorial points to help expedite the publication of your manuscript. Please direct any editorial questions to the journal office.

GENERAL GUIDELINES:

Text limits: Character count for a Report is < 20,000, not including spaces. Count includes title page, abstract, introduction, the joint Results & Discussion, and acknowledgments. Count does not include materials and methods, figure legends, references, tables, or supplemental legends.

Figures: Reports may have up to 5 main text figures. To avoid delays in production, figures must be prepared according to the policies outlined in our Instructions to Authors, under Data Presentation, <https://jcb.rupress.org/site/misc/ifora.xhtml>. All figures in accepted manuscripts will be screened prior to publication.

*****IMPORTANT:** It is JCB policy that if requested, original data images must be made available. Failure to provide original images upon request will result in unavoidable delays in publication. Please ensure that you have access to all original microscopy and blot data images before submitting your revision. ***

Supplemental information: There are strict limits on the allowable amount of supplemental data. Reports may have up to 3 supplemental figures. Up to 10 supplemental videos or flash animations are allowed. A summary of all supplemental material should appear at the end of the Materials and methods section.

Please note that JCB now requires authors to submit Source Data used to generate figures containing gels and Western blots with all revised manuscripts. This Source Data consists of fully uncropped and unprocessed images for each gel/blot displayed in the main and supplemental figures. Since your paper includes cropped gel and/or blot images, please be sure to provide one Source Data file for each figure that contains gels and/or blots along with your revised manuscript files. File names for Source Data figures should be alphanumeric without any spaces or special characters (i.e., SourceDataF#, where F# refers to the associated main figure number or SourceDataFS# for those associated with Supplementary figures). The lanes of the gels/blots should be labeled as they are in the associated figure, the place where cropping was applied should be marked (with a box), and molecular weight/size standards should be labeled wherever possible.

The typical timeframe for revisions is three to four months. While most universities and institutes have reopened labs and allowed researchers to begin working at nearly pre-pandemic levels, we at JCB realize that the lingering effects of the COVID-19 pandemic may still be impacting some aspects of your work, including the acquisition of equipment and reagents. Therefore, if you anticipate any difficulties in meeting this aforementioned revision time limit, please contact us and we can work with you to find an appropriate time frame for resubmission. Please note that papers are generally considered through only one revision cycle, so any revised manuscript will likely be either accepted or rejected.

Thank you for this interesting contribution to Journal of Cell Biology. You can contact us at the journal office with any questions, cellbio@rockefeller.edu or call (212) 327-8588.

Sincerely,

Johanna Ivaska
Monitoring Editor
Journal of Cell Biology

Tim Fessenden
Scientific Editor
Journal of Cell Biology

Reviewer #1 (Comments to the Authors (Required)):

Talins are the principal proteins responsible for connecting integrins to actin. This linkage is mechanosensitive, a process mediated by force-dependent unfolding of talin helical domains and the subsequent binding of other proteins. Talins interact with a range of adapters and enzymes, and they are currently considered the primary hub for regulated cytoskeletal association at integrin adhesion complexes.

Based on the outputs of a splice junction expression pipeline, which used RNA sequencing data, the authors identified a previously unannotated splicing event in the TLN1 gene (I assume this variant was not present in TLN2, but this is not commented on). Splice segment 17b inclusion results in a 17 aa extension of the linker between talin rod bundles 1 and 2, and predicted (AlphaFold) alterations in the structure of the linker-bound helices of both bundles. The authors find that 17b inclusion reduces the force required to unfold R1 and R2, and that in vitro this results in higher vinculin binding. Interestingly, 17b expression is tissue-specific and regulated by TGF β . In cells, 17b inclusion appears to increase focal adhesion number, but reduce their size.

As talin plays such a central role in adhesion, anything that modulates its function will be of interest to the field. In this context, phosphorylation and proteolytic cleavage have set precedents, and there is no doubt splicing would be a significant addition. The manuscript is submitted as a Report to JCB, which means a full characterization is not necessary; however, the evidence for a role for 17b in cells is currently weak and should be strengthened before publication.

Major comments

1. HeLa cells were used for the TGF β experiments. It would be better to employ another, more physiologically relevant, line to complement the data.

2. The adhesions shown in Fig. 5 are predominantly focal adhesions, and there is a relatively minor effect on number and size following 17b inclusion. Further data are needed to demonstrate that 17b has an effect in cells.

(a) The authors should comment in the methods on the relative levels of expression of talin variants. I assume they were equivalent.

(b) Nascent adhesions at the lamellipodial edge should be imaged and talin:vinculin ratios calculated. Is there an effect of 17b on the very earliest adhesions?

(c) Similarly, if the cells make fibrillar adhesions, is the focal to fibrillar transition altered by the presence of 17b (see Atherton et al., PMID 36074065).

(d) Are the kinetics of talin and/or vinculin in nascent adhesions or focal adhesions (as assessed by FRAP) altered by the

inclusion of 17b?

Minor comments

1. All capitalization of talin in the middle of sentences should be removed.
2. Calsyntenin is typically associated with synaptic junctions, but is more widely expressed. Is 17b localized to cell-cell adhesion junctions?

Reviewer #2 (Comments to the Authors (Required)):

This manuscript describes the identification of a talin-1 splice isoform. Using a recently published splicing analysis pipeline, the authors identify a talin-1 transcript that includes a 51 nucleotide-long exon leading to the insertion of 17 amino acids (aa) between the R1 and R2 domain of the talin rod. These analyses also reveal that the inclusion of the exon into the talin-1 transcript correlates with distinct drug response profiles of cancer cells and is modulated by TGF- β /SMAD3 signaling in HeLa cells. The biochemical analysis of the R1-R2 domain indicates that the insertion destabilizes the R1R2 structure, which unfolds at slightly lower forces when compared to the wildtype (WT) protein. Cell culture experiments indicate that expression of the newly discovered isoform affects focal adhesion (FA) formation leading to an increased number of FAs per cell. Overall, the identification of the new talin-1 isoform is interesting and both the splice isoform analysis as well as the biochemical characterization of the R1-R2 domain seem rather solid. The major limitation of the study is that it remains largely unclear whether and how the expression of the newly discovered splice isoform affects cellular function. While the in-silico data suggest a potential, pathophysiological function, the cell biological studies do not clarify in which context expression of the talin splice variant could become relevant. I therefore suggest to include additional cell biological experiments. Please see below for more detailed comments.

1.) To test how the expression of the talin splice isoform affects cells, the authors use talin-1 deficient cells that are reconstituted with WT-talin-1 or the newly discovered talin-1 splice variant. I find these experiments somewhat difficult to interpret as the utilized cell line still expresses talin-2 (see Kumar et al, JCB, 2016). Would the authors not expect to observe more obvious differences upon talin-2 depletion?

That being said, the current cell biological findings are, at this point, rather superficial. The main observation is that cells expressing the talin-1 splice variant "contained many more, smaller nascent adhesion". I believe that extending these experiments could significantly benefit the manuscript and significantly increase the impact of the study. Thus, I strongly encourage the authors to include the following experiments:

- Fig. 5B indicates the "number of FA per cell". This could be a consequence of cells being larger and, in fact, cells expressing the novel talin-1 splice variant appear slightly larger in Fig. 5A. The authors could quantify cell area (to calculate the FA/cell area) and confirm that the observed effect is not simply a result of an increased cell size.
- Another reason for a change in FA size and FA number could be that cells differ in their migratory behavior, and the talin-1 isoform expressing cells in Fig. 5A do seem to display more lamellipodia-type structures. As the authors well know, static cells often have larger FAs, while migratory cells typically display smaller, more dynamic FAs. The authors could easily address whether changes in FA number and size are caused by differential migratory behavior using live cell microscopy.
- An elegant, complementary experiment addressing both points mentioned above would be to analyze cells seeded onto micropatterned substrates (which are commercially available). Under these conditions, all cells would be in a static state and have roughly the same cell area. Are FA size and number still different under these conditions?
- Talin-1 is an important mediator between integrin receptors and the actin cytoskeleton, and since the talin-isoform expressing cells seem morphologically different (Fig. 5A), the authors should at least include a phalloidin staining to visualize the overall morphology of the actin network.
- The authors write in the main text that cells form more "nascent adhesions". As the authors will know, nascent adhesion are not just small FAs but adhesion structures that, for instance, form in a myosin-II independent fashion. However, evidence that talin-isoform expressing cells form more myosin-II independent adhesion structures is missing. The shown result demonstrate that FAs are smaller. Maybe the authors could clarify this point.

2.) The splicing analyses makes a number of interesting predictions. The impact of the study is somewhat limited by the fact that virtually none of these predictions are tested and it remains unclear how relevant the findings really are. However, such experiments would require much more work. As an alternative, I suggest that the authors make a much bigger effort to explain their data and contextualize their findings with the existing literature.

- It is thought that 92-94 % of all human genes are spliced. Given the size of the talin-1 gene, it is actually not too surprising that alternatively spliced talin-1 transcripts are observed and it could be expected that even additional splice isoforms exist. It is not clear from the manuscript whether the authors detected in any of the investigated data sets just one additional splice variant or more (Fig. 1B seems to suggest another skipping event that is especially prominent in BRCA samples). If additional splice variants were detected, I would like to encourage the authors to published these in this work. This would significantly broaden the scope of the study and potentially accelerate scientific progress. To the very least the authors should be very clear on whether only one or more alternative splice variants of talin-1 were detected.
- The text initially implies that the talin-17b splice variant is specifically expressed in cancer tissue. However, it is mentioned later

in the text that high expression of this splice variant is also observed in healthy tissue including skin, pancreas, kidney, liver, spleen, etc. I suggest that the data set in Supplementary Fig. 1B, showing the expression profile of the talin-17b transcript in normal tissue, is included in the main figure and more prominently featured.

- The authors report that the insertion of the 17aa insertion leads to a marked reduction in force required to unfold the R2 and R1 domains. While the WT-domain unfolds in response to forces of 15 pN and 25 pN, the R1R2-17b domain appears to unfold at 13 pN and 21 pN. This is an interesting observation, however, previous work by the authors indicated that talin is exposed to much lower forces in cells and they argued that talin-1 experiences force below 6 pN (Kumar et al., JCB, 2016). Data from another study seem to indicate that forces could be at or higher than 11 pN (Austen et al. NCB, 2015). Since an important prediction of the here presented data is that the insertion of the 17b insert leads to altered talin mechanics, it would be helpful to the reader if the existing literature on talin mechanics would be discussed in more detail. It is otherwise difficult to understand how a change from 15 pN to 13 pN could be relevant if the molecule is at no point exposed to force higher than 6 pN.

3.) In Fig. 3 an AlphaFold model is used to predict the structural changes caused by insertion of the additional 17 aa. Could the authors please indicate the confidence level of this prediction, for example by coloring the degree of confidence in the structure model?

4.) I believe Fig. 4B should be labelled differently. I apologize if I misunderstood but does the blue line not indicate "R1R2-WT + Vd1" instead of "R1R2-WT". Similarly, does the red line not indicate R1R2-17b + Vd1, instead of "R1R2-17b". If so, the labelling should be adjusted accordingly. In general, it would be easier to understand the data set if elution profiles of R1R2-WT and R1R2-17b without Vd1 were also shown. Could the authors also indicate the molecular weights associated with the different peaks?

5.) The NMR spectra seem to indicate that both R1R2-WT and R1R2-17b binds the RIAM peptide to about the same degree. Specifically, the authors write that chemical shifts were observed for the R2 domain. It would be helpful if the authors indicated the R2 peaks in the NMR spectra. It is not obvious to the non-NMR-expert, which peaks are specifically associated with R2.

Reviewer #1 (Comments to the Authors (Required)):

Talins are the principal proteins responsible for connecting integrins to actin. This linkage is mechanosensitive, a process mediated by force-dependent unfolding of talin helical domains and the subsequent binding of other proteins. Talins interact with a range of adapters and enzymes, and they are currently considered the primary hub for regulated cytoskeletal association at integrin adhesion complexes.

Based on the outputs of a splice junction expression pipeline, which used RNA sequencing data, the authors identified a previously unannotated splicing event in the TLN1 gene (I assume this variant was not present in TLN2, but this is not commented on). Splice segment 17b inclusion results in a 17 aa extension of the linker between talin rod bundles 1 and 2, and predicted (AlphaFold) alterations in the structure of the linker-bound helices of both bundles. The authors find that 17b inclusion reduces the force required to unfold R1 and R2, and that in vitro this results in higher vinculin binding. Interestingly, 17b expression is tissue-specific and regulated by TGF β . In cells, 17b inclusion appears to increase focal adhesion number, but reduce their size.

As talin plays such a central role in adhesion, anything that modulates its function will be of interest to the field. In this context, phosphorylation and proteolytic cleavage have set precedents, and there is no doubt splicing would be a significant addition. The manuscript is submitted as a Report to JCB, which means a full characterization is not necessary; however, the evidence for a role for 17b in cells is currently weak and should be strengthened before publication.

We thank the reviewer for the positive evaluation, and the constructive feedback. We have expanded the revised manuscript with more detailed characterisation of the cellular effects of 17b, and also of the TLN1/CLSTN1 concerted splicing program in multiple different cell types to demonstrate this link.

Major comments

1. HeLa cells were used for the TGF β experiments. It would be better to employ another, more physiologically relevant, line to complement the data.

The TGF- β analysis is using the RNA-Seq data from Tripathi *et al.* (Tripathi et al., 2016) who beautifully delineated the splicing pathway downstream of TGF- β (mediated via SMAD3 and PCBP1) in HeLa cells. However, the authors did not identify the novel TLN1 splice event, likely because their pipeline did not primarily focus on the discovery of non-annotated exons.

Our re-analysis of their RNA-Seq data using our new DJExpress pipeline enabled us to clearly visualise the concerted splicing program of TLN1 and CLSTN1.

Further, we now include new RT-qPCR data to show that we see this inverse reciprocity of the splicing of TLN1 and CLSTN1 in multiple cell lines and organoid models (**Fig. S1D**).

2. The adhesions shown in Fig. 5 are predominantly focal adhesions, and there is a relatively

minor effect on number and size following 17b inclusion. Further data are needed to demonstrate that 17b has an effect in cells.

We thank the reviewer for this suggestion upon which we have extensively expanded our analysis of 17b in cells as described below.

(a) The authors should comment in the methods on the relative levels of expression of talin variants. I assume they were equivalent.

The reviewer is correct, the expression levels of talin variants in the reconstitution experiments are equivalent. We now show this data in **Fig. S3B**.

(b) Nascent adhesions at the lamellipodial edge should be imaged and talin:vinculin ratios calculated. Is there an effect of 17b on the very earliest adhesions?

We thank the reviewer for raising this point and we have now carried out a time course analysing adhesions from 15 min to 4 hours, and calculated vinculin:talin ratios (**Fig. 5A,B**). The new results show that ratios are higher with talin 17b at both early and late times. We have also included a set where talin-2 was depleted, which slightly magnified differences in cell spreading but had no effect on vinculin:talin ratios.

(c) Similarly, if the cells make fibrillar adhesions, is the focal to fibrillar transition altered by the presence of 17b (see Atherton et al., PMID 36074065).

We have stained for tensin-1 to identify fibrillar adhesions (**Fig. 5I,J**). Results are similar for WT vs 17b talin.

(d) Are the kinetics of talin and/or vinculin in nascent adhesions or focal adhesions (as assessed by FRAP) altered by the inclusion of 17b?

Following the reviewer's suggestion, we have carried out photobleaching experiments, which showed no difference in rate of recovery for WT vs 17b talin (**Fig. 5G,H**) in focal adhesions. We did not extend this to small edge adhesions, in part because their small size and lower signal make analysis more difficult and in part because other analyses gave no reason to think that small edge adhesions revealed any novel aspect of talin 17b behaviour.

Minor comments

1. All capitalization of talin in the middle of sentences should be removed.

We apologize for this spelling mistake, and we have removed the capitalization of talin throughout the manuscript.

2. Calsyntenin is typically associated with synaptic junctions, but is more widely expressed.

The reviewer is correct, however, the role of Calystenin-1 in non-neuronal cells is not well understood, and future work will look at how and why CLSTN1 and TLN1 are coupled in this way. We now include our RT-qPCR data on the inverse splicing of CLSTN1 and TLN1 in multiple cell lines (**Fig. S1D**) that shows that the mutual exclusivity of these splice events is not limited to synaptic junctions but is more ubiquitous.

Is 17b localized to cell-cell adhesion junctions?

We assessed talin co-localisation with Beta-catenin to mark cell-cell contacts. We observed minimal co-localisation with either talin1 WT or 17b and no apparent difference between the two isoforms was observed (**Fig. S3C**)

We thank the reviewer for her/his interesting and valuable suggestions. We believe these new experiments have substantially strengthened the conclusions in our study.

Reviewer #2 (Comments to the Authors (Required)):

This manuscript describes the identification of a talin-1 splice isoform. Using a recently published splicing analysis pipeline, the authors identify a talin-1 transcript that includes a 51 nucleotide-long exon leading to the insertion of 17 amino acids (aa) between the R1 and R2 domain of the talin rod. These analyses also reveal that the inclusion of the exon into the talin-1 transcript correlates with distinct drug response profiles of cancer cells and is modulated by TGF- β /SMAD3 signaling in HeLa cells. The biochemical analysis of the R1-R2 domain indicates that the insertion destabilizes the R1R2 structure, which unfolds at slightly lower forces when compared to the wildtype (WT) protein. Cell culture experiments indicate that expression of the newly discovered isoform affects focal adhesion (FA) formation leading to an increased number of FAs per cell.

Overall, the identification of the new talin-1 isoform is interesting and both the splice isoform analysis as well as the biochemical characterization of the R1-R2 domain seem rather solid. The major limitation of the study is that it remains largely unclear whether and how the expression of the newly discovered splice isoform affects cellular function. While the in-silico data suggest a potential, pathophysiological function, the cell biological studies do not clarify in which context expression of the talin splice variant could become relevant. I therefore suggest to include additional cell biological experiments. Please see below for more detailed comments.

We thank the reviewer for the positive evaluation, and the useful feedback. We have expanded the cell biological experiments and made changes that we think improve the manuscript substantially.

1.) To test how the expression of the talin splice isoform affects cells, the authors use talin-1 deficient cells that are reconstituted with WT-talin-1 or the newly discovered talin-1 splice variant. I find these experiments somewhat difficult to interpret as the utilized cell line still expresses talin-2 (see Kumar et al, JCB, 2016). Would the authors not expect to observe more obvious differences upon talin-2 depletion?

We thank the reviewer for this thoughtful comment. We have now analysed cell spreading and focal adhesion characteristics with and without talin-2 knockdown. Depleting talin-2 slightly increases some of the differences between WT and 17b talin-1 but all of the general behaviours are similar (**Fig. 5A-F**).

That being said, the current cell biological findings are, at this point, rather superficial. The main observation is that cells expressing the talin-1 splice variant "contained many more, smaller nascent adhesions". I believe that extending these experiments could significantly benefit the manuscript and significantly increase the impact of the study. Thus, I strongly encourage the authors to include the following experiments:

- Fig. 5B indicates the "number of FA per cell". This could be a consequence of cells being larger and, in fact, cells expressing the novel talin-1 splice variant appear slightly larger in Fig. 5A. The authors could quantify cell area (to calculate the FA/cell area) and confirm that the observed effect is not simply a result of an increased cell size.

Thank you for the useful suggestions and the opportunity to improve the manuscript.

We have now determined the cell spread area is not different between WT and 17b talin.

- Another reason for a change in FA size and FA number could be that cells differ in their migratory behavior, and the talin-1 isoform expressing cells in Fig. 5A do seem to display more lamellipodia-type structures. As the authors well know, static cells often have larger FAs, while migratory cells typically display smaller, more dynamic FAs. The authors could easily address whether changes in FA number and size are caused by differential migratory behavior using live cell microscopy.

We have measured cell motility and found that TLN1 17b cells migrate significantly faster than TLN1 WT cells (**Fig. 5F**). Thus, we thank the reviewer for this suggestion as these data further support a functional role for TLN1 alternative splicing in cell physiology.

- An elegant, complementary experiment addressing both points mentioned above would be to analyze cells seeded onto micropatterned substrates (which are commercially available). Under these conditions, all cells would be in a static state and have roughly the same cell area. Are FA size and number still different under these conditions?

Thank you for this suggestion. While this technology will be interesting for future studies, given that there is no difference in cell area between WT and 17b, we feel that analysis of cells on micropatterned substrates is beyond the scope of this short report.

- Talin-1 is an important mediator between integrin receptors and the actin cytoskeleton, and since the talin-isoform expressing cells seem morphologically different (Fig. 5A), the authors should at least include a phalloidin staining to visualize the overall morphology of the actin network.

We have now added images of F-actin (**Fig. S3E**); overall actin intensity and morphology are not significantly different.

- The authors write in the main text that cells form more "nascent adhesions". As the authors will know, nascent adhesion are not just small FAs but adhesion structures that, for instance, form in a myosin-II independent fashion. However, evidence that talin-isoform expressing cells form more myosin-II independent adhesion structures is missing. The shown result demonstrate that FAs are smaller. Maybe the authors could clarify this point.

We agree with the reviewer that nascent adhesions are mechanistically distinct structures but this is not central to the conclusions of this manuscript. Thus, we have revised the text to refer to them as small edge adhesions vs larger focal adhesions.

2.) The splicing analyses makes a number of interesting predictions. The impact of the study is somewhat limited by the fact that virtually none of these predictions are tested and it remains unclear how relevant the findings really are. However, such experiments would require much more work. As an alternative, I suggest that the authors make a much bigger effort to explain their data and contextualize their findings with the existing literature.

- It is thought that 92-94 % of all human genes are spliced. Given the size of the talin-1 gene, it is actually not too surprising that alternatively spliced talin-1 transcripts are observed and it could be expected that even additional splice isoforms exist. It is not clear from the manuscript whether the authors detected in any of the investigated data sets just one additional splice variant or more (Fig. 1B seems to suggest another skipping event that is especially prominent in BRCA samples). If additional splice variants were detected, I would like to encourage the authors to published these in this work. This would significantly broaden the scope of the study and potentially accelerate scientific progress. To the very least

the authors should be very clear on whether only one or more alternative splice variants of talin-1 were detected.

We thank the reviewer for raising this interesting point. Talin-2, with its ~12 kb primary transcript length, has been reported to have several different alternative exons resulting in at least two productive protein coding isoforms according to ENSEMBL. These have been documented throughout the literature. Notably, TLN1 is significantly shorter (~8.6 kb length) and has no annotated alternative isoforms in public databases. Although an alternative exon in TLN1 might not seem surprising now that we have discovered it, it has been entirely missed for 40 years despite >2000 papers on talin in PubMed. We were only able to find it because of our DJExpress pipeline that we developed to specifically detect also non-annotated altered splice junctions. As a result of our finding, ENSEMBL confirmed our discovery of this new cassette exon and will include this information with the next genome assembly revision.

The key point to appreciate here is that this is a **natural splicing event** found in healthy tissues. Although we discovered it by comparing normal and cancer cell datasets, it is not mis-splicing due to a mutation that alters splicing.

As for additional splice events in talin-1 apart from exon 17b, we would like to emphasize that we looked at talin-1 junctions in many datasets from healthy and diseased cells across almost all tissues of the body. This extensive analysis across 1000s of samples detected only a single specific altered splice form of a cassette exon between exons 17 and 18. The peak in **Fig. 1B** the reviewer refers to does not reflect altered splicing, but suggests slightly increased read coverage of a constitutive junction, which anyway does not pass our FDR cut-off. Thus, this junction does not lead to any alternative splicing. Although our pipeline corrects for changes in gene expression, this junction might reflect slightly elevated expression levels of TLN1 in cancer tissues compared to healthy controls.

If there was an additional altered splicing in talin-1, we would almost certainly have detected it in at least one of these 1000s of samples, as our pipeline is designed specifically to detect these alterations.

- The text initially implies that the talin-17b splice variant is specifically expressed in cancer tissue. However, it is mentioned later in the text that high expression of this splice variant is also observed in healthy tissue including skin, pancreas, kidney, liver, spleen, etc. I suggest that the data set in Supplementary Fig. 1B, showing the expression profile of the talin-17b transcript in normal tissue, is included in the main figure and more prominently featured. Thank you for this excellent suggestion. We have now moved the expression profile of 17b in healthy tissues from Supplementary Fig.1B into the main text in **Fig. 1D**.

The exon was discovered by comparison of differential junctions between healthy and cancer cells. This analysis strategy needs some expression evidence in at least some of the control samples and would otherwise not be possible. But we have been careful not to claim cancer specificity, instead, we called it cancer-enriched. However, we have revised the text to emphasize that this is a natural splicing event that is often misregulated in cancer.

- The authors report that the insertion of the 17aa insertion leads to a marked reduction in force required to unfold the R2 and R1 domains. While the WT-domain unfolds in response to forces of 15 pN and 25 pN, the R1R2-17b domain appears to unfold at 13 pN and 21 pN. This is an interesting observation, however, previous work by the authors indicated that talin is exposed to much lower forces in cells and they argued that talin-1 experiences force below 6 pN (Kumar et al., JCB, 2016). Data from another study seem to indicate that forces could be at or higher than 11 pN (Austen et al. NCB, 2015). Since an important prediction of the here presented data is that the insertion of the 17b insert leads to altered talin mechanics, it would be helpful to the reader if the existing literature on talin mechanics would be discussed in more detail. It is otherwise difficult to understand how a change from 15 pN to 13 pN could be relevant if the molecule is at no point exposed to force higher than 6 pN. We appreciate the reviewer's point that this section needs clarification and have revised the discussion to make it easier to understand.

The key point here is that the Kumar study measured the average tension on talin which indeed is ~5 pN. The Austen study used a sensor that reports tension above 7-11 pN and found a *fraction* of talin experiences tension above this range. Analyses of tension on integrins reports spikes >20 pN (PMID 23704575). These transient high tension events likely trigger the unfolding events.

This is also clearly illustrated (PLoS Biol, 2011 PMID: 22205879) in which measurement of the end-to-end distance of talin in cells revealed cycles of stretch and relaxation changing length between 50-350 nm. In our 2016 study, Yao et al. (Nature Comms, 2016 PMID: 27384267) we analysed these end-to-end fluctuations and could simulate this response to force (Fig.6D,E in the Yao et al. paper). The average force on the talins was found to be <10 pN but with large spikes, which can be ~20 pN, and are what alters the talin switch patterns. As a talin domain unfolds it introduces slack into the linkage and so the tension on the molecule is lowered.

3.) In Fig. 3 an AlphaFold model is used to predict the structural changes caused by insertion of the additional 17 aa. Could the authors please indicate the confidence level of this prediction, for example by coloring the degree of confidence in the structure model?
We have now included the confidence level in **Fig. S1E**.

4.) I believe Fig. 4B should be labelled differently. I apologize if I misunderstood but does the blue line not indicate "R1R2-WT + Vd1" instead of "R1R2-WT". Similarly, does the red line not indicate R1R2-17b + Vd1, instead of "R1R2-17b". If so, the labelling should be adjusted accordingly.

Thank you for pointing this out, we have now amended the labelling of **Fig. 4B** to make it more clear.

In general, it would be easier to understand the data set if elution profiles of R1R2-WT and R1R2-17b without Vd1 were also shown.

We have now labelled the peaks in **Fig. 4B** to make it easier to follow. As requested we added the profiles of R1R2-WT and R1R2-17b alone in **Fig. S1F**. The R1R2-WT and VD1 elute at the same position (as can be seen in **Fig. 4B**), the R1R2-17b elutes slightly earlier due to it being slightly bigger.

Could the authors also indicate the molecular weights associated with the different peaks?

We added the MWs as calculated using our standard curve for the gel filtration column into the figure legend. The MW of the two complex peaks is calculated to be ~80 kDa and ~110 kDa which are 1:2 and 1:3 talin:vinculin complex peaks.

5.) The NMR spectra seem to indicate that both R1R2-WT and R1R2-17b binds the RIAM peptide to about the same degree.

This is correct, both R1R2-WT and R1R2-17b bind to the RIAM peptide. This is the purpose of this figure to show that the LD-motif binding site on R2 in the folded "0" state is intact in both isoforms.

Specifically, the authors write that chemical shifts were observed for the R2 domain. It would be helpful if the authors indicated the R2 peaks in the NMR spectra. It is not obvious to the non-NMR-expert, which peaks are specifically associated with R2.

We overlaid the separate spectra for R1 and R2 on top of the R1R2 spectrum in order to determine which peaks are from R1 and R2. The peaks in this region almost exclusively overlap those from R2 (see screenshot below). However, the complication here is that there are shift changes in R1R2 compared to the individual domains due to their interaction. Thus, some of the *individual* peaks cannot be unambiguously identified. Therefore, whilst we are confident that the data support the overall interpretation, we hesitate to label individual peaks R2.

Our previous study of RIAM binding to talin showed that R2 binds RIAM, which provides further support for this conclusion. However, if this is too speculative we can revise this statement.

We thank this reviewer for the thoughtful suggestions and ideas for improving our manuscript.

We thank all the reviewers for their helpful comments and suggestions, and we appreciate the enthusiasm they showed for our manuscript. We also would like to thank the reviewers for volunteering their time and effort to provide ideas for enhancing our manuscript. We think that the new experiments have substantially improved this manuscript. We hope that with these changes, the reviewers will find the manuscript acceptable for publication in the *Journal of Cell Biology*.

February 10, 2023

RE: JCB Manuscript #202209010R

Prof. Benjamin Thomas Goult
University of Kent
School of Biosciences,
I4/24 Ingram Building
Canterbury, Kent CT2 7NJ
United Kingdom

Dear Prof. Goult:

Thank you for submitting your revised manuscript entitled "A novel cancer-associated cassette exon in TLN1 alters talin-1 mechanosensitivity". We would be happy to publish your paper in JCB pending final revisions necessary to meet our formatting guidelines (see details below). While both reviewers recommend publication, Reviewer 2 remains concerned by the discrepancies and variation in talin-1 force measurements. We suggest modifying the discussion to better reflect these range of measurements reported here and in prior literature.

A. MANUSCRIPT ORGANIZATION AND FORMATTING:

Full guidelines are available on our Instructions for Authors page, <http://jcb.rupress.org/submission-guidelines#revised>. Submission of a paper that does not conform to JCB guidelines will delay the acceptance of your manuscript.

1) Text limits: Character count for Reports is < 20,000, not including spaces. Count includes abstract, introduction, results, discussion, and acknowledgments. Count does not include title page, figure legends, materials and methods, references, tables, or supplemental legends.

2) Figures limits: Reports may have up to five main figures and three supplemental figures/tables.

3) Figure formatting: Scale bars must be present on all microscopy images, including inset magnifications. Molecular weight or nucleic acid size markers must be included on all gel electrophoresis.

4) Statistical analysis: Error bars on graphic representations of numerical data must be clearly described in the figure legend. The number of independent data points (n) represented in a graph must be indicated in the legend. Statistical methods should be explained in full in the materials and methods. For figures presenting pooled data the statistical measure should be defined in the figure legends. Please also be sure to indicate the statistical tests used in each of your experiments (either in the figure legend itself or in a separate methods section) as well as the parameters of the test (for example, if you ran a t-test, please indicate if it was one- or two-sided, etc.). Also, if you used parametric tests, please indicate if the data distribution was tested for normality (and if so, how). If not, you must state something to the effect that "Data distribution was assumed to be normal but this was not formally tested."

** Please indicate statistical tests used and, if t-tests were used, whether they were one- or two-sided, in all figure legends.

5) Abstract and title: The abstract should be no longer than 160 words and should communicate the significance of the paper for a general audience. The title should be less than 100 characters including spaces. Make the title concise but accessible to a general readership.

** Because this work describes the identification of an existing cassette exon, we suggest changing the title to "TLN1 contains a cancer-associated cassette exon that alters talin-1 mechanosensitivity"

6) Materials and methods: Should be comprehensive and not simply reference a previous publication for details on how an experiment was performed. Please provide full descriptions in the text for readers who may not have access to referenced manuscripts.

** Please provide details on all methods beyond listing the relevant references.

7) Please be sure to provide the sequences for all of your primers/oligos and RNAi constructs in the materials and methods. You must also indicate in the methods the source, species, and catalog numbers (where appropriate) for all of your antibodies. Please also indicate the acquisition and quantification methods for immunoblotting/western blots.

8) Microscope image acquisition: The following information must be provided about the acquisition and processing of images:

a. Make and model of microscope

- b. Type, magnification, and numerical aperture of the objective lenses
- c. Temperature
- d. Imaging medium
- e. Fluorochromes
- f. Camera make and model
- g. Acquisition software
- h. Any software used for image processing subsequent to data acquisition. Please include details and types of operations involved (e.g., type of deconvolution, 3D reconstitutions, surface or volume rendering, gamma adjustments, etc.).

10) Supplemental materials: There are strict limits on the allowable amount of supplemental data. Reports may have up to 3 supplemental figures. Please also note that tables, like figures, should be provided as individual, editable files. A summary of all supplemental material should appear at the end of the Materials and methods section.

13) ORCID IDs: ORCID IDs are unique identifiers allowing researchers to create a record of their various scholarly contributions in a single place. At resubmission of your final files, please consider providing an ORCID ID for as many contributing authors as possible.

Please note that JCB now requires authors to submit Source Data used to generate figures containing gels and Western blots with all revised manuscripts. This Source Data consists of fully uncropped and unprocessed images for each gel/blot displayed in the main and supplemental figures. Since your paper includes cropped gel and/or blot images, please be sure to provide one Source Data file for each figure that contains gels and/or blots along with your revised manuscript files. File names for Source Data figures should be alphanumeric without any spaces or special characters (i.e., SourceDataF#, where F# refers to the associated main figure number or SourceDataFS# for those associated with Supplementary figures). The lanes of the gels/blots should be labeled as they are in the associated figure, the place where cropping was applied should be marked (with a box), and molecular weight/size standards should be labeled wherever possible.

WHEN APPROPRIATE: The source code for all custom computational methods published in JCB must be made freely available as supplemental material hosted at www.jcb.org. Please contact the JCB Editorial Office to find out how to submit your custom macros, code for custom algorithms, etc. Generally, these are provided as raw code in a .txt file or as other file types in a .zip file. Please also include a one-sentence summary of each file in the Online Supplemental Material paragraph of your manuscript.

B. FINAL FILES:

-- Cover images: If you have any striking images related to this story, we would be happy to consider them for inclusion on the journal cover. Submitted images may also be chosen for highlighting on the journal table of contents or JCB homepage carousel.

Images should be uploaded as TIFF or EPS files and must be at least 300 dpi resolution.

****It is JCB policy that if requested, original data images must be made available to the editors. Failure to provide original images upon request will result in unavoidable delays in publication. Please ensure that you have access to all original data images prior to final submission.****

****The license to publish form must be signed before your manuscript can be sent to production. A link to the electronic license to publish form will be sent to the corresponding author only. Please take a moment to check your funder requirements before choosing the appropriate license.****

Thank you for this interesting contribution, we look forward to publishing your paper in Journal of Cell Biology.

Sincerely,

Johanna Ivaska
Monitoring Editor
Journal of Cell Biology

Tim Fessenden
Scientific Editor
Journal of Cell Biology

Reviewer #1 (Comments to the Authors (Required)):

None

Reviewer #2 (Comments to the Authors (Required)):

The authors have made a very good job in addressing the comments; the newly included data sets and adjustments have further improved the manuscript. I would like to congratulate the authors on a very interesting piece of work and recommend the publication of this study.

I have no further suggestions except for one more comment on the discussion of talin mechanics. I apologize if overlooked this, but I did not find any mention of an average tension of 5 pN in the Kumar et al paper. This makes sense, I believe, because the Kumar et al study uses a biosensor that is sensitive to forces of 1-6 pN. If a fraction of talin molecules is experiencing higher forces up to 20 pN, then a reliable estimation of an average tension is simply not possible.

In any case, it may not be very helpful assigning an average tension value to talin as the molecule is exposed to high and low forces, depending on its subcellular location and the cell biological/physiological context. Maybe the authors would consider to slightly change the discussion accordingly.